# Comparison of Aircraft Measurements during GoAmazon2014/5 and ACRIDICON-CHUVA

Fan Mei[1], Jian Wang[2], Jennifer M. Comstock[1], Ralf Weigel[13], Martina Krämer[14], Christoph Mahnke[13, 8], John E. Shilling[1], Johannes Schneider[8], Charles N. Long[7], Manfred Wendisch[5], Luiz A. T. Machado[3], Beat Schmid[1], Trismono Krisna[5], Mikhail Pekour[1], John Hubbe[1], Andreas Giez[6], Bernadett Weinzierl[6], Martin Zoeger[6], Christiane Schulz[8], Mira L. Pöhlker[8], Hans Schlager[6], Micael A. Cecchini[9], Meinrat O. Andreae[8,10], Scot T. Martin[4], Suzane, S. de Sá[4], Jiwen Fan[1], Jason Tomlinson[1], Stephen Springston[2], Ulrich Pöschl[8], Paulo Artaxo[11], Christopher Pöhlker[8], Thomas Klimach[8] , Andreas Minikin[12], Armin Afchine[14], Stephan Borrmann[13,8]

1. Pacific Northwest National Laboratory, Richland, WA, United States.
2. Brookhaven National Laboratory, Upton, NY, United States.
3. National Institute for Space Research (INPE), São Paulo, Brazil
4. Harvard University, Cambridge, MA, United States
5. University of Leipzig, Leipzig, Germany
6. Deutsches Zentrum für Luft- und Raumfahrt (DLR), Oberpfaffenhofen, Germany
7. NOAA ESRL GMD/CIRES, Boulder, CO, United States
8. Max Planck Institute for Chemistry, Mainz, Germany
9. University of São Paulo (USP), São Paulo, Brazil
10. Scripps Institution of Oceanography, University of California San Diego, La Jolla, California, USA
11. Instituto de Física, Universidade de São Paulo, São Paulo, Brazil
12. DLR Oberpfaffenhofen, Flight Experiments Facility, Wessling, Germany
13. Institute for Physics of the Atmosphere, Johannes Gutenberg University, Mainz, Germany
14. Research Centre Jülich, Institute for Energy and Climate Research 7: Stratosphere (IEK-7), Jülich, Germany

*Correspondence to:* Fan Mei (fan.mei@pnnl.gov)

**Abstract.** The indirect effect of atmospheric aerosol particles on the Earth's radiation balance remains one of the most uncertain components affecting climate change throughout the industrial period. This issue is partially a result of the incomplete understanding of aerosol-cloud interactions. One objective of the GoAmazon2014/5 and ACRIDICON-CHUVA projects was to improve the understanding of the influence of the emissions of the tropical megacity of Manaus

(Brazil) on the surrounding atmospheric environment of the rainforest and to investigate its role in the life cycle of convective clouds. During one of the intensive observation periods (IOPs) in the dry season from September 1 to October 10, 2014, comprehensive instrument suites collected data from several ground sites. In a coordinated way, the advanced suites of sophisticated instruments were deployed in situ both from the U.S. Department of Energy Gulfstream-1 (G1) aircraft and the German High Altitude and Long-Range Research Aircraft (HALO) during three coordinated flights on September 9, 21, and October 1. Here we report on the comparison of measurements collected by the two aircraft during these three flights. Such comparisons are difficult to obtain, but they are essential for assessing the data quality from the individual platforms and quantifying their uncertainty sources. Similar instruments mounted on the G1 and HALO collected vertical profile measurements of aerosol particles number concentration and size distribution, cloud condensation nuclei concentration, ozone, and carbon monoxide concentration, cloud droplet size distribution, and downward solar irradiance. We find that the above measurements from the two aircraft agreed within the range given by the measurement uncertainties. Aerosol chemical composition measured by instruments on HALO agreed with the corresponding G1 data collected at high altitudes only. Furthermore, possible causes of discrepancies between the data sets collected by the G1 and HALO instrumentation are addressed in this paper. Based on these results, criteria for meaningful aircraft measurement comparisons are discussed.

## 1. Introduction

Dominated by biogenic sources, the Amazon basin is one of the few remaining continental regions where atmospheric conditions realistically represent those of the pristine or pre-industrial era (Andreae et al., 2015). As a natural atmospheric "chamber", the area around the urban region of Manaus in central Amazonia is an ideal location for studying the atmosphere under natural conditions as well as under conditions influenced by human activities and biomass burning events (Andreae et al., 2015; Artaxo et al., 2013; Davidson et al., 2012; Keller et al., 2009; Kuhn et al., 2010; Martin et al., 2016b; Pöhlker et al., 2018; Poschl et al., 2010; Salati and Vose, 1984). The Observations and Modeling of the Green Ocean Amazon (GoAmazon2014/5) campaign was conducted in 2014 and 2015 (Martin et al., 2017; Martin et al., 2016b). The primary objective of GoAmazon2014/5 was to improve the quantitative understanding of the effects of anthropogenic

influences on atmospheric chemistry and aerosol-cloud interactions in the tropical rainforest area. During the dry season in 2014, the ACRIDICON (Aerosol, Cloud, Precipitation, and Radiation Interactions and Dynamics of Convective Cloud Systems)-CHUVA (Cloud Processes of the Main Precipitation Systems in Brazil) campaign also took place to study tropical convective clouds and precipitation over Amazonia (Wendisch et al., 2016).

A feature of the GoAmazon 2014/5 field campaign was the design of the ground sites' location, which uses principles of Lagrangian sampling to align the sites with the Manaus pollution plume (Figure 1: Source location – Manaus (T1 site), and downwind location – Manacapuru (T3 site)). The ground sites were overflown with the low-altitude U.S. Department of Energy (DOE) Gulfstream-1 (G1) aircraft and the German High Altitude and Long Range Research Aircraft (HALO). These two aircraft are among the most advanced in atmospheric research, deploying suites of sophisticated and well-calibrated instruments (Schmid et al., 2014; Wendisch et al., 2016). The pollution plume from Manaus was intensively sampled during the G1 and HALO flights and also by the DOE Atmospheric Radiation Measurement (ARM) program Mobile Aerosol Observing System and ARM Mobile Facility located at one of the downwind surface sites (T3 site- 70 km west of Manaus). The routine ground measurements with coordinated and intensive observations from both aircraft provided an extensive data set of multi-dimensional observations in the region, which serves i) to improve the scientific understanding of the influence of the emissions of the tropical megacity of Manaus (Brazil) on the surrounding atmospheric environment of the rainforest and ii) to understand the life cycle of deep convective clouds and study open questions related to their influence on the atmospheric energy budget and hydrological cycle.

As more and more data sets are merged to link the ground-based measurements with aircraft observations, and as more studies focus on the spatial variation and temporal evolution of the atmospheric properties, it is critical to quantify the uncertainties ranges when combining the data collected from the different platforms. Due to the challenges of airborne operations, especially when two aircraft are involved in data collection in the same area, direct comparison studies are rare. However, this type of study is critical for further combining the datasets between the ground sites and aircraft. Thus, the main objectives of the study herein are to demonstrate how to achieve meaningful comparisons between two moving platforms, to conduct detailed comparisons between data collected by two aircraft, to identify the potential measurement issues, to quantify

reasonable uncertainty ranges of the extensive collection of measurements, and to evaluate the measurement sensitivities to the temporal and spatial variance. The comparisons and the related uncertainty estimations quantify the current measurement limits, which provide realistic measurement ranges to climate models as initial conditions to evaluate their output.

The combined GoAmazon2014/5 and ACRIDICON-CHUVA field campaigns not only provide critical measurements of aerosol and cloud properties in an under-sampled geographic region but also provide a unique opportunity to understand and quantify the quality of these measurements using closely orchestrated comparison flights. The comparisons between the measurements from similar instruments on the two research aircraft can be used to identify potential measurement issures and quantify the uncertainty range of the field measurements, which include primary meteorological variables (Section 3.1), trace gases concentrations (Section 3.2), aerosol particle properties (number concentration, size distribution, chemical composition, and microphysical properties) (Section 3.3), cloud properties (Section 3.4), and downward solar irradiance (Section 3.5). We evaluate the consistency between the measurements aboard the two aircraft for a nearly full set of gas, aerosol particle, and cloud variables. Results from this comparison study provide the foundation not only for assessing and interpreting the observations from multiple platforms (from the ground to low altitude, and then to high altitude) but also for providing high-quality data to improve the understanding of the accuracy of the measurements related to the effects of human activities in Manaus on local air quality, terrestrial ecosystems in rainforest, and tropical weather.

## 2. Measurements

### 2.1 Instruments

The ARM Aerial Facility deployed several in situ instruments on the G1 to measure atmospheric state parameters, trace gases concentrations, aerosol particle properties, and cloud characteristics (Martin et al., 2016b; Schmid et al., 2014). The instruments installed on HALO covered measurements of meteorological, chemical, microphysical, and radiation parameters. Details of measurements aboard HALO are discussed in the ACRIDICON-CHUVA campaign overview paper (Wendisch et al., 2016). The measurements compared between the G1 and HALO are listed in Table 1.

#### 2.1.1 Atmospheric parameters

All G1 and HALO meteorological sensors were routinely calibrated to maintain measurement accuracy. The G1 primary meteorological data were provided at one-second time resolution based on the standard developed by the Inter-Agency Working Group for Airborne Data and Telemetry Systems (Webster and Freudinger, 2018). For static temperature measurement, the uncertainty given by the manufacturer (Emerson) is ±0.1 K, and the uncertainty of the field data is ±0.5 K. The static pressure had a measurement uncertainty of 0.5 hPa. The standard measurement uncertainties were ±2 K for the chilled mirror hygrometer and 0.5 $ms^{-1}$ for wind speed.

On HALO, primary meteorological data were obtained from the Basic HALO Measurement and Sensor System (BAHAMAS) at one-second time resolution. The system acquired data from airflow and thermodynamic sensors and from the aircraft avionics and a high-precision inertial reference system to derive the basic meteorological parameters like pressure, temperature, the 3D wind vector, aircraft position, and attitude. Water vapor mixing ratio and further derived humidity quantities were measured by the Sophisticated Hygrometer for Atmospheric Research (SHARC) based on direct absorption measurement by a tunable diode laser (TDL) system. The absolute accuracy of the primary meteorological data was 0.5 K for air temperature, 0.3 hPa for air pressure, 0.4-0.6 $ms^{-1}$ for wind, and 5% (±1 ppm) for water vapor mixing ratio. All sensors were routinely calibrated and traceable to national standards (Giez et al., 2017; Krautstrunk and Giez, 2012).

### 2.1.2 Gas phase

Constrained by data availability, this comparison of trace gas measurements is focused on carbon monoxide (CO) and ozone ($O_3$) concentrations. Those measurements were made aboard the G1 by a $CO/N_2O/H_2O$ instrument (Los Gatos Integrated Cavity Output Spectroscopy instrument model 907-0015-0001), and an Ozone Analyzer (Thermo Scientific, Model 49i), respectively. The G1 CO analyzer was calibrated for response daily by NIST-traceable commercial standards before the flight. Due to the difference between laboratory and field conditions, the uncertainty of CO measurement is about ±5% for one-second sampling periods, An ultra-fast carbon monoxide monitor (Aero Laser GmbH, AL5002) was deployed on HALO. The detection of CO is based on a vacuum-ultraviolet-fluorimetry, employing the excitation of CO at 150 nm and the precision is 2 ppb, and the accuracy is about 5%. The ozone analyzer measures ozone concentration based on the absorbance of ultraviolet light at a wavelength of 254 nm. The ozone analyzer (Thermo Scientific, Model 49c) in the HALO payload is very similar to the one on the

G1 (Model 49i), with an accuracy greater than 2 ppb or about ±5% for four-second sampling periods. The G1 ozone monitor was calibrated at the New York State Department of Environmental Conservation testing laboratory at Albany.

### 2.1.3 Aerosol

Aerosol number concentration was measured by different condensation particle counters (CPCs) on the G1 (TSI, CPC 3010) and HALO (Grimm, CPC model 5.410). Although two CPCs were from different manufacturers, they were designed using the same principle, which is to detect particles by condensing butanol vapor on the particles to grow them to a large enough size that they can be counted optically. Both CPCs were routinely calibrated in the lab and reported the data at one-second time resolution. The HALO CPC operated at 0.6-1 L min$^{-1}$, with a nominal cutoff of 4 nm. Due to inlet losses, the effective cutoff diameter increases to 9.2 nm at 1000 hPa, and 11.2 nm at 500 hPa (Andreae et al., 2018; Petzold et al., 2011). The G1 CPC operated at 1 L min$^{-1}$ volumetric flow rate and the nominal cut-off diameter $D_{50}$ measured in the lab was ~10 nm. During a flight, the cut-off diameter may vary due to tubing losses, which contribute less than 10 % uncertainty to the comparison between two CPC concentrations.

Two instruments deployed on the G1 measured aerosol particle size distribution. a Fast Integrated Mobility Spectrometer (FIMS) inside of the G1 cabin measured the aerosol mobility size from 15 to 400 nm (Kulkarni and Wang, 2006a, b; Olfert et al., 2008; Wang, 2009). The ambient aerosol particles were charged after entering the FIMS inlet and then separated into different trajectories in an electric field based on their electrical mobility. The spatially separated particles grow into super-micrometer droplets in a condenser where supersaturation of the working fluid is generated by cooling. At the exit of the condenser, a high-speed charge-coupled device camera captures the image of an illuminated grown droplet at high resolution. In this study, we used the FIMS 1 Hz data for comparison. The size distribution data from FIMS were smoothed. Aside from the FIMS, The airborne version of the Ultra High Sensitivity Aerosol Spectrometer (UHSAS) was deployed on G1 and HALO. The G1 and HALO UHSAS were manufactured by the same company, and both were mounted under the wing on a pylon. UHSAS is an optical-scattering, laser-based particle spectrometer system. The size resolution is around 5% of the particle size. The G1 UHSAS typically covered a size range of 60 nm to 1000 nm. HALO UHSAS covered 90 nm to 500 nm size range for the September 9 flight.

Based on operating principles, FIMS measures aerosol electrical mobility size and UHSAS measures aerosol optical equivalent size. Thus, the difference in the averaged size distributions from those two types of instruments might be linked to differences in their underlying operating principles, such as the assumption in the optical properties of aerosol particles. The data processing in the G1 UHSAS assumed that the particle refractive index is similar to ammonium sulfate (1.55), which is larger than the average refractive index (1.41-0.013i) from a previous Amazon study (Guyon et al., 2003). The HALO UHSAS was calibrated with polystyrene latex spheres, which has a refractive index about 1.572 for the UHSAS wavelength of 1054 nm. The uncertainty due to the refraction index can lead to up to 10% variation in UHSAS measured size (Kupc et al., 2018). Also, the assumption of spherical particles affects the accuracy of UHSAS sizing of ambient aerosols.

The chemical composition of submicron non-refractory ($NR\text{-}PM_1$) organic and inorganic (sulfate, nitrate, ammonium) aerosol particles was measured using a high-resolution time-of-flight aerosol mass spectrometer (HR-ToF-AMS) aboard the G1 (DeCarlo et al., 2006; Jayne et al., 2000; Shilling et al., 2018; Shilling et al., 2013). Based on the standard deviation of observed aerosol mass loadings during filter measurements, the HR-ToF-AMS detection limits for the average time of thirteen seconds are approximately 0.13, 0.01, 0.02, 0.01 (3σ values) $\mu g\ m^{-3}$ for organic, sulfate, nitrate, and ammonium, respectively (DeCarlo et al., 2006). A Compact Time-of-flight Aerosol Mass Spectrometer (C-ToF-AMS) was operated aboard HALO to investigate the aerosol composition. Aerosol particles enter both the C-ToF-AMS and HR-ToF-AMS via constant pressure inlets controlling the volumetric flow into the instrument, although the designs of the inlets are somewhat different (Bahreini et al., 2008). The details about the C-ToF-AMS operation and data analysis are reported in Schulz's paper (Schulz et al., 2018). The overall accuracy has been reported as ~30 % for both AMS instruments (Alfarra et al., 2004; Middlebrook et al., 2012). Data presented in this section were converted to the same condition as the HALO AMS data, which is 995 hPa and 300 K.

The number concentration of cloud condensation nuclei (CCN) was measured aboard both aircraft using the same type of CCN counter from Droplet Measurement Technologies (DMT, model 200). This CCN counter contains two continuous-flow, thermal-gradient diffusion chambers for measuring aerosols that can be activated at constant supersaturation. The supersaturation is created by taking advantage of the different diffusion rates between water vapor

and heat. After the supersaturated water vapor condenses on the CCN in the sample air, droplets are formed, counted and sized by an Optical Particle Counter (OPC). The sampling frequency is one second for both deployed CCN counters. Both CCN counters were calibrated using ammonium sulfate aerosol particles in the diameter range of 20-200 nm. The uncertainty of the effective water vapor supersaturation was ±5%. (Rose et al., 2008)

### 2.1.4 Clouds

Aircraft-based measurements are an essential method for in situ sampling of cloud properties (Brenguier et al., 2013; Wendisch and Brenguier, 2013). Over the last 50–60 years, hot-wire probes have been the most commonly used devices to estimate liquid water content (LWC) in the cloud from research aircraft. Since the 1970s, the most widely used technique for cloud droplet spectra measurements has been developed based on the light-scattering effect. This type of instrument provides the cloud droplet size distribution as the primary measurement. By integrating the cloud droplet size distribution, additional information, such as LWC can be derived from the high-order data product.

Three cloud probes from the G1 were discussed in this manuscript. The Cloud Droplet Probe (CDP) is a compact, lightweight forward-scattering cloud particle spectrometer that measures cloud droplets in the 2 to 50 μm size range (Faber et al., 2018). Using a state-of-the-art electro-optics and electronics, Stratton Park Engineering (SPEC Inc.) developed a Fast Cloud Droplet Probe (FCDP), which also use forward-scattering to determine cloud droplet distributions and concentrations in the same range as CDP with up to 100 Hz sampling rate. The G1 also carried a two-dimensional stereo probe (2DS, SPEC Inc.), which has two 128-photodiode linear arrays working independently and electronics produce shadowgraph images with 10 μm pixel resolution. Two orthogonal laser beams cross in the middle of the sample volume, with the sample cross section for each optical path of 0.8 $cm^2$. The manufacturer claims the maximum detection size is up to 3000 μm for the 2DS. However, due to the counting statistic issue, the data used in this study is from 10–1000 μm only (Lawson et al., 2006). 2DS was upgraded with modified probe tips, and an arrival time algorithm was applied to the 2DS data processing. Both efforts effectively reduced the number of small (shattered) particles (Lawson, 2011). For G1 cloud probes, the laboratory calibrations of the sample area and droplet sizing were performed before the field deployment.

During the deployment, biweekly calibrations with glass beads were performed with the size variation of less than 5%, which were consistent with the pre-campaign and after-campaign calibrations. Comparison between the LWC derived from cloud droplet spectra with hot-wire LWC measurement was made to estimate/eliminate the coincidence errors in cloud droplet concentration measurements (Lance et al., 2010; Wendisch et al., 1996)

On board of HALO, two cloud probes were operated and discussed in this manuscript, each consisting of a combination of two instruments: the Cloud Combination Probe (CCP) and a Cloud Aerosol Precipitation Spectrometer (CAPS, denoted as NIXE-CAPS; NIXE: Novel Ice Experiment). The CCP is a combination of a CDP (denoted as CCP-CDP) with a CIPgs (Cloud Imaging Probe with grey scale, DMT, denoted as CCP-CIPgs). NIXE-CAPS consists of a CAS-Dpol (Cloud and Aerosol Spectrometer, DMT, denoted as NIXE-CAS) and a CIPgs (denoted as NIXE-CIPgs). CIPgs is an optical array probe comparable to the 2DS operated on the G1. CIPgs obtains images of cloud elements using a 64-element photodiode array (15μm resolution) to generate two-dimensional images with nominal detection diameter size range from 15 to 960 μm (Klingebiel et al., 2015; Molleker et al., 2014).The CCP-CDP detects the forward-scattered laser light by cloud particles in size range of 2.5 to 46 μm. The sample area of the CCP- CDP was determined to be 0.27±0.025 $mm^2$ with an uncertainty of less than 10% (Klingebiel et al., 2015). CAS-Dpol (or NIXE-CAS) is a light scattering probe comparable to the CDP but covers the size range of 0.6 to 50 μm in diameter, thus including the upper size range of the aerosol particle size spectrum (Luebke et al., 2016). Furthermore, CAS-Dpol measures the polarization state of the particles (Costa et al., 2017). Correspondingly to the G1 CDP, the performance of the CCP-CDP and NIXE-CAS were frequently proven by glass beads calibrations. Prior to or after each HALO flight, CCP-CIPgs and NIXE-CIPgs calibrations were performed by using a mainly transparent spinning disc that carries opaque spots of different but known size. The data of the CCP measured particle concentration on board of HALO are corrected to gain ambient conditions using a thermodynamic approach developed by (Weigel et al., 2016). For NIXE-CAPS, the size distributions were provided where NIXE-CAS was merged with the NIXE-CIPgs at 20 μm.

### 2.1.5 Solar radiation

The G1 radiation suite included shortwave (SW, 400 - 2,700 nm) broadband total upward and downward irradiance measurements using Delta-T Devices model SPN-1 radiometers. The

radiation data were corrected for aircraft tilt from the horizontal reference plane. A methodology has been developed (Long; et al., 2010) for using measurements of total and diffuse shortwave irradiance and corresponding aircraft navigation data (latitude, longitude, pitch, roll, heading) to calculate and apply a correction for platform tilt to the broadband hemispheric downward SW measurements. Additionally, whatever angular offset there may be between the actual orientation of each radiometer's detector and what the navigation data say is level has also been determined for the most accurate tilt correction.

The Spectral Modular Airborne Radiation measurement sysTem (SMART-Albedometer) was installed aboard HALO. Depending on the scientific objective and the configuration, the optical inlets determining the measured radiative quantities can be chosen. The SMART-Albedometer has been utilized to measure the spectral upward and downward irradiances; thereby it is called as an albedometer, as well as to measure the spectral upward radiance. The SMART-Albedometer is designed initially to cover measurements in the solar spectral range between 300 and 2,200 nm (Krisna et al., 2018; Wendisch et al., 2001; Wendisch et al., 2016). However, due to decreasing sensitivity of the spectrometers at large wavelengths, the use of the wavelengths was restricted to 300 – 1,800 nm. The spectral resolution is defined by the full width at half maximum (FWHM), which is between 2 and 10 nm. In this case, the instruments were mounted on an active horizontal stabilization system for keeping the horizontal position of the optical inlets during aircraft movements (up to +/- 6 degrees from the horizontal plane).

### 2.2 Flight patterns

During the dry season IOP (September 1 – October 10, 2014), two types of coordinated flights were carried out: one flight in cloud-free condition (September 9) and two flights with clouds present (September 21 and October 1). In this study, we compare the measurements for both coordinated flight patterns. The discussion is mainly focused on the flights under cloud-free conditions on September 9 and the flight with clouds present on September 21, as shown in figure 1. The other coordinated flight on October 1 is included in the supplemental document.

For the cloud-free coordinated flight, the G1 took off first and orbited around an area from the planned rendezvous point until HALO arrived in sight. It then coordinated with HALO and performed a wing-to-wing maneuver along straight legs around 500 m above sea level, as shown in Figure 2. The normal G1 average sampling speed is 100 m $s^{-1}$, and the normal HALO average

sampling speed is 200 m $s^{-1}$. During the coordinated flight on September 9, both aircraft also adjusted their normal sampling speed by about 50 m $s^{-1}$ so that they could fly side by side.

For the second type of coordinated flights, the G1 and HALO flew the stacked paten at their own normal airspeed. On September 21, the G1 also took off from the airport first, followed by HALO 15 minutes later. Then, both aircraft flew above the T3 ground site and subsequently flew several flight legs stacked at different altitudes. The two aircraft were vertically separated by about 330 m and sampled below, inside, and above clouds. Due to the different aircraft speeds, the flight distance between two aircraft flight paths continued increasing from 15 min to 1 hour, as shown in Figure 3. On October 1, the G1 focused on the cloud microphysical properties and contrasting polluted versus clean clouds. HALO devoted the flight to the cloud vertical evolution and life cycle and also probed the cloud processing of aerosol particles and trace gases. The G1 and HALO coordinated two flight legs between 950–1250 m above the T3 site under cloud-free conditions. Following that, HALO flew to the south of Amazonia, and the G1 continued sampling plume-influenced clouds above the T3 site, and then flew above the Rio Negro area.

In this study, to perform a meaningful comparison of in situ measurements, all the data from instruments were time synchronized with the aircraft (G1 or HALO) navigation system. For AMS and CPC data, the time shifting due to tubing length and instrument flow had been corrected. For the coordinated flight on Sep. 9, the data compared was from the same type of measurements with the same sampling rate. For the measurements with the different sampling rate, the data were binned to the same time interval for comparison. For the flight with the cloud presents (Sep.21 and Oct. 1), the following criteria are used: 1) the data collected by the two aircraft must be less than 30 mins apart from each other; 2) the comparison data was binned to 200 m altitude interval; and 3) the cloud flag was applied to the aerosol measurements, and the data affected by the cloud shattering are eliminated from the comparisons of aerosol measurements. Moreover, additional comparison criteria are specified for individual measurements in the following section. Tables 2 shows the total number of points used for the comparison.

# 3. Results

## 3.1 Comparison of the G1 and HALO measurements of atmospheric state parameters

The atmospheric state parameters comprise primary variables observed by the research aircraft. The measurements provide essential meteorological information not only for understanding the

atmospheric conditions but also for providing the sampling conditions for other measurements, such as those of aerosol particles, trace gases, and cloud microphysical properties.

For cloud-free coordinated flights, the comparison focused on the near side-by-side flight leg at around 500 m, as shown in Figure 2. Table 3 shows the basic statistics of the data for primary atmospheric state parameters. In general, the atmospheric state parameters observed from both aircraft were in excellent agreement. The linear regression achieved a slope was near 1 for four individual measurements. The regression is evaluated using the below equation 1.

$$R^2 = 1 - \frac{SS_{regression}}{SS_{Total}} \quad (1)$$

Where the sum squared regression error is calculated by $SS_{regression} = \sum(y_i - y_{regression})^2$, and the sum squared total error is calculated by $SS_{Total} = \sum(y_i - \bar{y})^2$, $y_i$ is the individual data point, $\bar{y}$ is the mean value, and $y_{regression}$ is the regression value. When the majority of the data points are in a narrow value range, using the mean is better than the regression line and the $R^2$ will be negative (Neg in Table 3).

The difference between the average ambient temperatures on the two aircraft was 0.5 K, and the difference between the average dew point temperatures was about 1 K. For temperature and humidity, the G1 data were slightly higher than the HALO data. The main contributions to the observed differences include the error propagation in the derivation of the ambient temperature from the measured temperature, instrumental-measurement uncertainty, and the temporal and spatial variability. The average horizontal wind speed measured by HALO is 0.4 m $s^{-1}$ higher than the average horizontal wind speed measured by the G1. The uncertainty source of wind estimation is mainly due to the error propagation from the indicated aircraft speed measurement and the aircraft ground speed estimation from GPS. The static pressure distribution measured aboard HALO showed a smaller standard deviation (0.9 hPa) compared to the value of the G1 (1.5 hPa). The standard deviation (std) was also 0.6 hPa narrower. Part of the reason for this difference is a more substantial variation of the G1 altitude during level flight legs when G1 flew at around 50 m/s higher than its normal airspeed. Thus, any biases caused by their near side-by-side airspeeds being different from their normal airspeeds would be undetected during these coordinated flights.

For the coordinated flights under cloudy conditions, we used the criteria from Section 2 to compare ambient conditions measured by the G1 and HALO aircraft. As shown in Figure 4, the linear regression slopes for ambient temperature, dew point temperature, and pressure were also

close to 1 between the G1 and HALO measurements during the September 21 coordinated flight. The $R^2$ value is also close to 1. These results suggest that the G1 and HALO measurements achieved excellent agreements. The dew point temperature from the G1 measurement was erroneous between 2200–2700 m and above 3700 m (Figure 4(c)) because the G1 sensor was skewed by wetting in the cloud. The HALO dew point temperature was calculated from the total water mixing ratio measured by TDL, and that measurement in the cloud was more accurate than the measurement made by the chilled mirror hydrometer aboard the G1.

The lower value of the $R^2$ value in horizontal wind speed means the ratio of the regression error and total error in wind measurement is much higher than the temperature and pressure measurements. The main contributions to this difference are the error propagation during the horizontal wind speed estimation and also the temporal and spatial variance between two aircraft sampling location. We observed differences between the two aircraft data of up to 2 m $s^{-1}$, caused by the increasing sampling distance as the two aircraft were climbing up. For example, the G1 flew a level leg above T3 around 2500 m between 16:20-16:30, while HALO stayed around 2500 m for a short period and kept climbing to a higher altitude. Due to strong vertical motion, turbulence, and different saturations (evaporation-condensation processes), the variances in the horizontal wind speed (Figure 4(d)) were also more significant compared to the variances of temperature and pressure measurements.

### 3.2 Comparison of trace gas measurements

For the cloud-free coordinated flight, ozone is the only trace gas measurement available on both aircraft. The linear regression slope shows that the HALO ozone concentration was about 8% higher than the G1 concentration. The difference between the averaged ozone concentrations was 4.1 ppb. The primary source of bias is probably the different ozone loss in the sampling and transfer lines.

The comparison made on September 21 flight in Figure 5 shows good agreement for the vertically averaged ozone measurements. Comparing the statistics data from September 9, the ozone measurement is not sensitive to the temporal and spatial changes. The G1 CO measurement shows a low $R^2$ value which is mainly from the systematic bias between the two instruments with different operation principles. The larger variance between 2000-3000 m in CO concentration

indicates the spatial variation contribution, while the G1 and HALO were sampling different air masses, as indicated in Figure S7.

### 3.3 Comparison of aerosol measurements

Aerosol particles exhibited strong spatial variations, both vertically and horizontally, due to complex atmospheric processing in the Amazon basin, especially with the local anthropogenic sources at Manaus. Thus, any spatially resolved measurement is critical to characterizing the properties of the Amazonian aerosols. The cloud-free coordinated flights provide us with suitable data to compare the G1 and HALO aerosol measurements and thus will enable further studies combined with the ground measurements. The vertical profiles obtained using the G1 and HALO platforms in different aerosol regimes in the Amazon basin have contributed to many studies (Fan et al., 2018; Martin et al., 2017; Wang et al., 2016).

When comparing the measurements from the two aircraft, the inlet system is a critical item, especially for sampling aerosol particles (Wendisch et al., 2004). Inlet design and characterization can actively modify the measured aerosol particle number concentration, size distribution, and chemical composition. The G1 aerosol inlet is a fully automated isokinetic inlet. Based on the manufacturer wind tunnel test and peer-reviewed publications, this inlet operates for aerosol up to 5 μm, with transmission efficiency around 50 % at 1.5 μm (Dolgos and Martins, 2014; Kleinman et al., 2007; Zaveri et al., 2010). The HALO sub-micrometer Aerosol Inlet (HASI) was explicitly designed for HALO. Based on the numerical flow modeling, optical particle counter measurements, and field study evaluation, HASI has a cut-off size of 3 μm, with transmission efficiency larger than 90 % at 1 μm (Andreae et al., 2018; Minikin et al., 2017).

#### 3.3.1 Aerosol particle number concentration

For the cloud-free coordinated flight, the linear regression of CPC and UHSAS between the G1 and HALO measurements are also included in Table 3. Also, the total number concentration of HALO CPC data was about 20 % lower than the total number concentration from the G1 CPC, as shown in Figure 6. The typical uncertainty between two CPCs is 5-10%.(Gunthe et al., 2009; Liu and Pui, 1974) Many factors can contribute to the rest 10-15% difference. Those include systematic uncertainties, such as systematic instrument drifts, different aerosol particle losses inside of the two CPCs, and different inlet transmission efficiencies in the two aircraft.

The CPC data in Figure 6 are color-coded with UTC time. The general trend is that the aerosol number concentration increased with aircraft sampling through the Manaus plume between 15:30-15:40. A similar trend was observed in aerosol particle number concentration (Figure 7) measured by the Ultra-High Sensitivity Aerosol Spectrometer (UHSAS)-Airborne version (referred to as UHSAS). The total number concentration data given by UHSAS (Figure 7) is integrated over the overlapping size range (90 – 500 nm for the September 9 flight) for both the G1 and HALO UHSAS. The linear regression shows that the total aerosol particle concentration from HALO UHSAS is about 16.5% higher than the total aerosol concentration from the G1 UHSAS. The discrepancy between the two UHSAS measurements is mainly due to the error propagation in the sampling flow, the differential pressure transducer reading, the instrument stability, and calibration repeatability, which is consistent with the other UHSAS study (Kupc et al., 2018). In the airborne version of UHSAS, mechanical vibrations have a more significant impact on the pressure transducer reading than the case for the bench version of UHSAS.

For the coordinated flight on September 21, the G1 and HALO data are averaged to a 200 m vertical altitude interval, as shown in Figure 8. There was a good agreement in the CPC comparison, especially at altitudes of 200 m and above (<10 % variance). However, the linear regression slope significantly decreased, which was primarily related to the temporal and spatial differences in aerosol number concentration. Especially between 2000 – 3000 m altitude, the difference between the G1 and HALO measurement is largely due to the different aerosol srouces, as shown in Figure S6(a). The size range was integrated from 100–700 nm for UHSAS on September 21. The change of the size range was because the overlap of size distribution from both UHSAS instruments was changed. The linear regression slope and the $R^2$ value slightly decreased in the UHSAS comparison as shown in Figure 8(b). And we can expect that the main contribution to the UHSAS measurement difference is from instrument systematic drift and the spatial/temporal variance of particle concentration in the ambient environment as shown in Figure S6(b). However, combining with the CPC measurement comparison, it indicates that the smaller size aerosol particles (< 100 nm) have a more profound variance due to the temporal and spatial change.

### 3.3.2 Aerosol particle size distribution

For the cloud-free coordinated flight, the averaged aerosol size distribution measured during one flight leg is compared in Figure 9. Based on the comparison plot, at the size range less than 90

nm, the G1 UHSAS overestimated the particle counts, which is due to the error in the counting efficiency correction. The UHSAS detection efficiency is close to 100% for particles larger than 100 nm and concentrations below 3000 $cm^{-3}$ but decreases both for smaller particles and for higher concentrations considerably (Cai et al., 2008). The aerosol counting efficiency correction determined from the lab condition does not represent the real correction during the flight operation. Between 90 nm and 250 nm, FIMS agreed well with the G1 UHSAS, whereas HALO UHSAS is about 30 % higher than the other two instruments. For the size range of 250–500 nm, FIMS had good agreement with HALO UHSAS, whereas FIMS is about 30-50 % higher than the G1 UHSAS depending on the particle size. Because the UHSAS has a simplified “passive” inlet, the large size aerosol particle loss in the UHSAS inlet was expected to increase with the increase of the aircraft speed. Thus, the lower G1 UHSAS counts at a larger aerosol particle size are likely related to the particle loss correction.

For the September 21 flight, the aerosol size distribution vertical profiles were averaged into 100 m altitude intervals (Figure 10). Overall, all size distribution measurements captured the mode near 100 nm between 800–1000 m, which is at the top of the convective boundary layer, as indicated by the potential temperature (Figure 10(d)), which starts from a maximum near the ground and then becomes remarkably uniform across the convective boundary layer. With the increase in altitude, we observe the peak of the aerosol size distribution shifted from 100 nm to 150 nm. Note that due to data availability, the aerosol size distribution data from the HALO UHSAS has a less spatial resolution.

#### 3.3.3 A significant contribution of small aerosol particles

Comprehensive characterization of aerosol particles, especially small ones (<50 nm) during GoAmazon2014/5 has demonstrated that high concentrations of those small particles in the lower free troposphere are transported from the free troposphere into the boundary layer by a strong convective downdraft and sustain the population of particles in the pristine Amazon boundary layer. This important observation improved the current understanding of the aerosol influence on cloud properties and climate under natural conditions (Fan et al., 2018; Wang et al., 2016). However, the aerosol particle size distribution measurement, especially for sizes less than 50 nm, is very rare due to the lack of high-frequency airborne measurements. The most common aerosol size spectrometer, UHSAS, covers aerosol particle sizes larger than 60 nm. The scanning

mobility particle spectrometer cannot obtain size distribution in 1 Hz time resolution. The other approach is to estimate particle size distribution by extrapolating the UHSAS or Passive Cavity Aerosol Spectrometer Probe (size range 100 – 3000 nm) measured aerosol size distribution to smaller size ranges (down to 10 nm). The accuracy of the third approach is limited by the nature of the aerosol size distribution, and the aerosol particle concentration can be significantly underestimated if there is a dominant nucleation mode in the aerosol particle size distribution, such as during a new particle formation event.

As shown in Figure 11, we compared the integrated aerosol number concentrations between one wet season flight (on March 7), which was influenced by a long-range transport plume from Africa (Moran-Zuloaga et al., 2018) and one typical dry season flight (September 21). The agreement of the small aerosol number concentration between the FIMS-measured size distribution and UHSAS/PCASP estimated size distribution is reasonably good for the dry season flight when the accumulation mode dominated the aerosol particle size distribution. During the wet season, there was a strong vertical gradient in the particle size spectrum above central Amazonia under clean conditions. Thus, we can observe an increase of underestimation of the small size particle concentration both for the size ranges less than 50 nm and less than 100 nm, as the filled markers move away from each other with the increase of altitude. Because of the negligible mass contribution to the total aerosol loading, those ultrafine aerosol particles (< 50 nm) are conventionally considered too small to affect cloud formation. However, the new observational evidence and numerical simulation of deep convective clouds outlined a new mechanism, which suggests an energetic anthropogenic invigoration of deep convective clouds by those ultrafine aerosol particles in previously pristine regions of the world (Fan et al., 2018). Two newly published studies (Fan et al., 2018; Wang et al., 2016) emphasize the importance of the airborne observation and suggest the ultrafine aerosol particles (<50 nm) measurement should be included as a baseline routine measurement in future airborne experiments.

For field studies without the deployment of FIMS, one option to assess the accuracy of UHSAS/PCASP estimated size distribution is to compare the total number concentration based on the integration of the UHSAS/PCASP estimated size distribution to the total number concentration from CPC. For field study focusing on the high concentration and variability of sub-50 or sub-100 nm aerosol particles, such as new particle formation events, it is highly recommended to request

the deployment of FIMS. Due to the limited availability of FIMS, one option is to use several well-characterized CPCs, which operate at the different cut-off sizes, to measure the ambient aerosol simultaneously, and then use the data-inversion technique to estimate the aerosol size distribution of sub-50 or sub-100 nm aerosol particles. Another reasonable substitute to the FIMS might be a Scanning Mobile Particle Sizer (SMPS), but it should be noted that on an airborne platform an SMPS does not nearly have the same time resolution as a FIMS. To better adapt the spatial change in aerosol concentration, a residence chamber similar to a system described in another study (Kotchenruther and Hobbs, 1998) should be deployed with SMPS.

#### 3.3.4 Aerosol particle chemical composition

Figure 12(a) shows vertical profiles of the total mass concentrations measured by the two AMS instruments on September 21. Above 2500 m altitude, the agreement between the two instruments is excellent (mean difference less than 5%). Between 2000 and 2500 m, the agreement is within the uncertainty range. Below 2000 m altitude, however, the aerosol particle mass concentrations measured by the AMS operated on HALO are lower than the concentrations measured by the AMS on the G1. The aerosol volume concentrations from G1 AMS was converted from the mass concentration from AMS, by assuming the organic compound density was 1.5 g cm-3 (Pöschl et al., 2010). The converted aerosol volume concentration agreed well with the volume concentration calculated based on UHSAS data below 2500 m, as shown in Figure 12(b). The agreement at lower altitudes suggests that the lower concentration in HALO AMS is due to the transmission efficiency issue in the constant pressure inlet used by HALO AMS. This inlet was a prototype, designed and built at MPIC Mainz, and works by changing the size of the critical orifice that regulates the flow into the aerodynamic lens. The design and transmission characteristics will be described in an upcoming publication (Molleker, S., in prep.). The AMS aboard the G-1 used a constant pressure inlet based on the design in Bahreini et al., 2008. Thus, we conclude that data above 2500 m altitude measured by the AMS aboard HALO in 2014 are valid, while data below 2500 m need to be corrected using correction factors derived from laboratory characterization before further study. After 2014, the HALO inlet design was improved to address the inlet transmission issues specific to this field campaign.

#### 3.3.5 CCN number concentration

These measurements provide valuable information about the aerosol's ability to form cloud droplets and modify the microphysical properties of clouds. Numerous laboratory and field studies have improved the current understanding of the connections between aerosol particle size, chemical composition, mixing states and CCN activation properties (Bhattu and Tripathi, 2015; Broekhuizen et al., 2006b; Chang et al., 2010; Duplissy et al., 2008; Lambe et al., 2011; Mei et al., 2013a; Mei et al., 2013b; Pöhlker et al., 2016; Thalman et al., 2017). In addition, based on the simplified chemical composition and internal mixing state assumption, various CCN closure studies have achieved success within ±20% uncertainty for ambient aerosols (Broekhuizen et al., 2006a; Mei et al., 2013b; Rissler et al., 2004; Wang et al., 2008).

According to earlier studies (Gunthe et al., 2009; Pöhlker et al., 2016; Roberts et al., 2001; Roberts et al., 2002; Thalman et al., 2017), the hygroscopicity ($\kappa_{CCN}$) of CCN in the Amazon basin is usually dominated by organic components ($\kappa_{Org}$). Long-term ground-based measurements at the Amazon Tall Tower Observatory also suggested that there were low temporal variability and no pronounced diurnal cycles in hygroscopicity only under natural rainforest background conditions (Pöhlker et al., 2018; Pöhlker et al., 2016).

Using FIMS and CCN data from both the G1 and HALO collected during the coordinated flight leg on September 9, the critical dry diameter ($D_{50}$) was determined by integrating FIMS size distribution to match the CCN total number concentration. Then, $D_{50}$ was combined with the CCN-operated supersaturation to derive the effective particle hygroscopicity by applying the k-Köhler theory. The histogram plots based on the density of the estimated hygroscopicity ($\kappa_{est}$) from both aircraft were compared for the flight leg above T3. For the G1 and HALO data, the $\kappa_{est}$ value derived from the flight leg above the T3 site is 0.186±0.067 and 0.189±0.083 separately. That value is also slightly higher than the overall mean kappa derived from long-term measurements from the Amazon Tall Tower Observatory, which is 0.17±0.06 (Pöhlker et al., 2016; Thalman et al., 2017).

An example of a comparison of vertical profiles of the CCN concentration at 0.5% supersaturation on September 21 is shown in Figure 13. The difference between CCN measurements on the two aircraft is about 20% on average. However, the linear regression slope increases to 0.9120 if we focused on the data above 2500 m. The main contributions to the

difference include the difference in aerosol inlet structure, aerosol particle loss correction in the main aircraft inlet and the constant pressure inlet, the systematic inlet difference below 2500 m as shown in AMS data, and the error propagation of CCN measurements.

### 3.4 Comparison of cloud measurements

In situ cloud measurements help to capture the diversity of different cloud forms and their natural temporal and spatial variability. The G1 CDP and FCDP were deployed under the different wing pylons, and also on the different side of the aircraft. The G1 2DS was deployed on the same side of FCDP. The HALO cloud combination probe (CCP-CDP and CCP-CIPgs) and NIXE-CAPS (NIXE-CAS and NIXE-CIPgs) were deployed under the different wing pylons but on the same side of the aircraft. On September 21, 2014, based on the aircraft location and elevation information as shown in Figure 1(b) and Figure 3, two aircraft were sampling above T3 site and passing through the same cloud field at ~1600 m flight leg and ~1900 m flight leg as shown in Figure S8 and Figure S9. We used the cloud probes data from ~1900 m flight leg for the cloud droplet number concentration comparison. Two size ranges were considered: 3-20 μm from light scattering probes (CDP vs. FCDP on the G1, CCP-CDP vs. NIXE-CAS on HALO) and 2-960 μm from combined cloud probes.

#### 3.4.1 Comparison of cloud droplet number concentration between 3-20 μm

The primary cloud layer was observed by both the G1 and HALO between 1000-2500 m above ground. Although the two aircraft have sampled along the same flight path, the instruments probably observed different sets of the cloud due to cloud movement with the prevailing wind or different cloud evolution stages. Thus, an initial comparison focuses on the redundant instruments on the same aircraft, that measured truly collocated and synchronous on board of HALO and of the G1, respectively. In Figure 14 (a), the data of the CCP-CDP and of the NIXE-CAS are juxtaposed sampled over about 13 minutes for particles detection size ranges which were considered as most equivalent. The comparison reveals two ranges of particle number concentrations at which densification of agreeing measurements become visible. At very low number concentrations (about $10^{-1}$ – 10 per cm³) the presence of inactivated (interstitial) aerosols in the clear air space between the very few cloud elements should be considered. Over specific ranges, however, the fine structure of varying cloud droplet number concentration may cause the

regression's scattering, indicated by cloud particle measured by one instrument whilst respective antagonist seems to measure within almost clear air – and vice versa. At higher number concentrations, i.e., between $10^2$ and $10^3$ per $cm^3$, the comparison of the highly resolved data constitutes increasing compactness with respect to the 1:1 line. The overall data scatter of this comparison, however, may indicate the highly variable structure within clouds as those investigated over the Amazon basin. The data of the G1 CDP and the FCDP are juxtaposed as the same as HALO cloud probes. However, the sampled cloud period was much shorter – about 3 minutes. Similar as the HALO cloud probes comparison, we observe two ranges of particle number concentrations at which a densification of agreeing measurements become visible, especially for the lower number concentrations, in Figure 14(b). At higher number concentrations, only a few cloud elements were observed by the G1 cloud probes. That is because the G1 was about 7-23 minutes later to pass the same location as HALO, and experienced much fewer cloud elements.

#### 3.4.2 Comparison of cloud droplet size distribution between 2-960 μm from both aircraft

Comparing the cloud probes from the G1 and HALO, although it seems that the size distributions never match better than for the cloud particle diameter size range below 10 μm, the size distributions from two aircraft are in remarkably good agreement, considering the instance that the cloud detection on board the G1 occurred 7-23 minutes after the cloud probing on board of HALO, as shown in Figure 15. On HALO, the CCP and NIXE-CAPS probes agreed very well between 2-960 μm and both peaked around 10 μm. On the G1, although CDP and FCDP has a more significant difference in the size range less than 8 μm, both of them showed the peak of the size distribution was around 15 μm. The difference between the G1 CDP and FCDP may be due to the data post-processing. Additional coincidence correction and shattering correction were applied to FCDP, but not to CDP. For cloud elements larger than 10 μm, the difference between the obtained cloud particle size distributions from two aircraft becomes substantial (up to two orders of magnitude) which may be indicative for observations of two different stages within the progressing development of a precipitation cloud which is particularly expressed in elevated number concentrations of larger cloud elements observed during the G1 measurement that happened later. We also observed that the general cloud characteristic is similar at different altitude levels as shown in Figure S10. The first two of three averaged periods were chosen during the flight leg of ~1600 m and the last average period is for the flight leg ~1900 m compared in Figure

15. Due to the averaging, the fine in-cloud structure gets suppressed. The small scale variabilities inside a cloud which are illustrated by the scattering of the highly resolved measurement data from the instrument comparison (cf. Figure 14) and the temporal evolution of in-cloud microphysics are not ascertainable and furthermore are beyond the scope of this study.

### 3.5 Comparison of radiation measurements

In this study, the downward irradiance measured by the SPN-1 unshaded center detector was compared with the integrated downward irradiance from the SMART-Albedometer between 300–1,800 nm wavelengths in Figure 16. Only measurements from flight legs, where the G1 and HALO flew near side-by-side and at the same altitude were taken into consideration for analysis. In Figure 16, the top panel shows the time series of SPN-1 measurements, and the bottom panel shows the time series of SMART-Albedometer measurements. The black dots represented all data, and the blue circles identified data when the navigation condition was within +/- 1 degree from the horizontal level. The large scatter in the data between 15:12-15:28 and 15:35-15:40 is mainly due to the different sensor trajectories during the maneuvering of the aircraft to get to the coordinated flight position. Because of the difference of each aircraft position from horizontal, the measured signal varied from the signal of the direct component of sunlight. Each sensor might look at different directions of the sky or different parts of the clouds. In addition, both aircraft flew under scattered clouds, and this uneven sunlight blocking is another contribution to the "drop-off" behavior in the time series plots of the downward irradiance.

Comparing the G1 and HALO measurements between 15:15-1:55 using the restricted navigation criteria in Figure 17, we observed that the G1 SPN1 irradiance is slightly higher than the integrated irradiance from the SMART-Albedometer. However, the difference in the averaged irradiance is less than 10 %. That result could be due to the difference in radiometer spectral ranges: 400–2700 nm (SPN1 radiometer) vs. 300–1800 nm (SMART).

## 4 Uncertainty assessment

As mentioned in the introduction, a low-flying G1 and a high-flying HALO cover the sampling area from the atmospheric boundary layer, low clouds to the free troposphere, and the sampling period from the dry and wet seasons (Martin et al., 2016a). This spatial coverage provides the user community with abundant atmospheric-related data sets for their further studies, such as for remote

sensing validation and modeling evaluation. However, one critical step to bridge the proper usage of the observation with further atmospheric science study is to understand the measurement uncertainty in this data set, especially the variation between the coexisting measurements due to the temporal and spatial difference.

For the major measurements during this field study, three primary sources contribute to the measurement variation between the two aircraft: the temporal and spatial variations, the difference in the inlet characterization, and the limitation of the instrument capability. The difference in the inlet characterization and the instrument error are the same between the coordinated flights on September 9 and September 21. Thus, we can examine the sensitivity of each measurement to the spatial variation by comparing two flights. For the majority of the comparisons of the September 21 flight, there are no significant spatial and temporal variation between two aircraft measurements. However, we noticed the comparison uncertainty is more significant between 2000 – 3000 m altitude than the other altitudes in the aerosol and trace gas profile, especially for the aerosol particles smaller than 100 nm. This additional difference occurring between 2000-3000 m indicates the spatial variation contribution, while the G1 and HALO were sampling different air masses. The G1 had one flight leg around 2500 m above T3 site, while HALO continued climbing through 2000-3000 m range to reach the next flight leg around 4500 m. Thus, for the G1 measurements, the data show two modes in the histogram distribution. The large mode was typically from the data when the G1 passed through the pollution plume, and the small mode represented the background value. Because the flight path of HALO did not pass through the plume, their data shows only one mode in the histogram plots, as shown in S6 and S7 in the supplemental material.

For atmospheric meteorological variables, the overall uncertainty is relatively smaller (less than 1 %) comparing to the other airborne measurements. The main contribution for the three-dimensional wind measurement is more sensitive to the spatial variation than the ambient temperature and pressure due to the complex turbulence structure in the boundary layer (see Figure S5). The other measurement affected mainly by the spatial and temporal variation is the cloud measurements, which is consistent with a previous study (Andreae et al., 2004). The considerable variation in the comparison between 2-960 μm indicates the evolution of cloud droplet size

distributions (DSDs) over time and space has a more significant influence on the large droplet size, and it serves as the major contributor for the DSDs comparison.

The inlet also significantly affects the aerosol and gas phase measurements. Inlet characterizations are inherently challenging. However, comparisons as shown here can be used to assess the performance of the inlets indirectly. In this study, reasonable agreement of the total number concentration of aerosol particles between two CPCs indicated the uncertainty caused by the main aerosol inlet difference is less than 15%. In addition to the main aerosol inlet, the particle losses caused by AMS aerosol constant pressure inlet also affects the AMS comparison below 2500 m altitude. Based on a literature survey, this study, for the first time, compares the non-refractory particle mass concentration between two aircraft measurements. Although two AMS sampled different air masses during the majority of the campaign, the excellent agreement between the two measurements from the comparison flight linked the aerosol chemical composition from the wet to dry season and from the atmospheric boundary layer to the upper troposphere.

We also noticed that the CCN and UHSAS comparisons are associated with more substantial uncertainties because of the more complex instrument designs. The aerosol flow fluctuation, the CCN column temperature fluctuation, and the stabilization of the optical particle counter all contribute to the accurate estimation of the CCN concentration. In a similar sense, the aerosol flow fluctuation, the difference in the inlet efficiency at different platform speeds, the laser temperature fluctuation, and the signal-to-noise ratio at lower size range all contribute to the considerable uncertainty of the UHSAS concentration measurement. However, the CCN hygroscopicity estimation on the near side-by-side comparison on September 9, 2014, shows very encouraging agreement. Thus, the spatial variance and the instability of the CCN and UHSAS performance both led to the variance between two aircraft of up to 50% based on the comparison scenario on September 21, 2014. This remains the most significant variance we observed during these two aircraft measurement comparisons.

The summary of the major measurement uncertainty contributed by the spatial difference between the two aircraft is listed in Table 4.

## 5 Summary

In situ measurements made by well-characterized instruments installed on two research aircraft (the G1 and HALO) during the GoAmazon 2014/5 and ACRIDICON-CHUVA campaigns were compared. Overall, the analysis shows good agreement between the G1 and HALO measurements for a relatively broad range of atmospheric-related variables in a challenging strophic environment. Measured variables included atmospheric state parameters, aerosol particles, trace gases, clouds, and radiation properties. This study outlines the well-designed coordinated flights for achieving a meaningful comparison between two moving platforms. The high data quality was ensured by the most sophisticated instruments aboard two aircraft used the most advanced techniques, assisted with the best-calibrated/characterized procedures. The comparisons and the related uncertainty estimations quantify the current measurement limits, which provide the guidance to the modeler to realistically quantify the modeling input value and evaluate the variation between the measurement and the model output. The comparison also identified the measurement issues, outlined the associated reasonable measurement ranges, and evaluated the measurement sensitivities to the temporal and spatial variance.

The comparisons presented here were mainly from two coordinated flights. The flight on September 9 was classified as a cloud-free flight. During this flight, the G1 and HALO flew near side-by-side within a “polluted” leg, which was above the T3 site and across the downwind pollution plume from Manaus, and a “background” leg, which was outbound from Manaus to the west and could be influenced by the regional biomass burning events during the dry season. Both legs were at 500 m altitude and showed linear regression slopes of ambient temperature and pressure, horizontal wind speed and dew point temperature near to 1 between the G1 and HALO measurements. These comparisons provide a solid foundation for further evaluation of aerosol, trace gas, cloud, and radiation properties. The total aerosol concentration from CPC and UHSAS were compared for the 500 m flight leg above the T3 site. The UHSAS measurement had a better agreement than the CPC measurement. That is because of the minor difference in the inlet structure and instrument design between two UHSAS aboard the two aircraft. The average size distribution from both UHSAS and one FIMS in the G1 suggests that UHSAS had an over-counting issue at the size range between 60-90 nm, which was probably due to electrical noise and small signal-to-noise ratio in that size range. Good agreement in the aerosol size distribution measurement provides a “sanity” check for AMS measurements. A CCN closure study suggested that FIMS

provides valuable size coverage for better CCN number concentration estimation. Based on the $\kappa$-Köhler parameterization, $\kappa_{eff}$ observed at 500 m above the T3 site is 0.18±0.09, which is similar to the overall mean kappa from long-term ATTO measurements - 0.17±0.06 (Pöhlker et al., 2016). This similarity suggests that there is no significant spatial variability along the downwind transect, although the freshly emitted aerosol particles may have much less hygroscopicity. The difference in the ozone measurement comparison is about 4.1 ppb, which suggests that the bias due to the sampling line loss inside of the G1 gas inlet. The irradiance from the SPN1 unshaded center detector in the G1 was compared with the HALO integrated downward irradiance between 300–1800 nm and achieved a very encouraging agreement with a variance of less than 10%.

During the second type of the coordinated flights on September 21 (with cloudy conditions), HALO followed the G1 after take-off from Manus airport; then the two aircraft flew stacked legs relative to each other at different altitudes above the T3 site. For atmospheric state parameters, nearly linear correlations between the G1 and HALO were observed for ambient pressure, temperature, and dew point temperature measurements at an altitude range from ground to around 5000 m. Cloud presence affected the measurements of dew point temperature in the G1, resulting in a large discrepancy in the dew point temperature measurement and the derived relative humidity between 2000–3000 m. The horizontal wind had more variation than the rest of the meteorological properties, which is mainly due to the temporal and spatial variability. The aerosol number concentration comparison had an excellent agreement (<15 %) for aerosol particles larger than 10 nm counted by the CPC below 2500 m. While the integrated aerosol number concentration from UHSAS showed consistent discrepancy at different altitudes, that suggests the significant temporal and spatial variation of smaller aerosol particles (<100 nm). Although the aircraft-based UHSAS is a challenging instrument to operate, a reasonable size distribution profile comparison was made between both UHSAS and FIMS in the G1. The aerosol concentration measured by AMS instruments showed a mean difference less than 5% above 2500 m during the flight on September 21, although due to the ongoing study, the correction factors allowing for correction of data below 2500 m are not available yet. The difference between CCN number concentration measured on the two aircraft was on average 20%, and these data show the same altitude behavior as the AMS data. The main contributions to this difference include the difference in aerosol inlet structure, the aerosol loss correction in a constant pressure inlet, and the error propagation of CCN measurements. The ozone and CO vertical profile comparisons show a variation of less than 10%.

The ozone measurement variation is mainly from the systematic bias between two instruments with different operation principles, especially at altitudes higher than 4000 m.

Cloud probe comparisons were made for the cloud droplet number concentration between 3–20 µm for the initial comparison between the redundant instruments on the same aircraft. Then the comparison of cloud droplet size distribution between 2-960 µm for a flight leg around 1900 m showed a remarkably good agreement. The major cloud appearance was captured by both aircraft, although the cloud elements observed were affected by the cloud movement with the prevailing wind and the different cloud evolution stages. Furthermore, the relatively short time delay of 7-23 minutes between the independent measurements may give a hint for the time scales in which the cloud droplet spectra develop within a convective cloud over the Amazon basin.

The above results provide additional information about the reasonableness of measurements for each atmospheric variable. This study confirms the high-quality spatial and temporal dataset with clearly identified uncertainty ranges had been collected from two aircraft and builds a good foundation for further studies on the remote sensing validation and the spatial and temporal evaluation of modeling representation of the atmospheric processing and evolution.

**Acknowledgments**: This study was supported by the U.S. DOE, Office of Science, Atmospheric System Research Program, and used data from Atmospheric Radiation Measurement Climate Research Aerial Facility, a DOE Office of Science User Facility. The Pacific Northwest National Laboratory (PNNL) is operated for DOE by Battelle under contract DE-AC05-76RL01830. This work was also supported by the Max Planck Society, the DFG (Deutsche Forschungsgemeinschaft, German Research Foundation) HALO Priority Program SPP 1294, the German Aerospace Center (DLR), the FAPESP (São Paulo Research Foundation) Grants 2009/15235-8 and 2013/05014-0, and a wide range of other institutional partners. The contributions from Micael A. Cecchini were funded by FAPESP grant number 2017/04654-6.

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

Table 1. List of compared measurements and corresponding instruments deployed aboard the G1 and HALO during GoAmazon2014/5. The acronyms are defined in a table at the end of this paper. Dp indicates the particle diameter. □Dp refers to the size resolution.

| Measurement Variables | Instruments deployed on the G1 (Martin et al., 2016b; Schmid et al., 2014) | Instruments deployed on HALO (Wendisch et al., 2016) |
|---|---|---|
| Static Pressure | Rosemount (1201F1), 0-1400 hPa | Instrumented nose boom tray (DLR development), 0-1400 hPa |
| Static air temperature | Rosemount E102AL/510BF<br><br>-50 to +50 °C | Total Air Temperature (TAT) inlet (Goodrich/Rosemount type 102) with an open wire resistance temperature sensor (PT100),<br>-70 to +50 °C |
| Dewpoint temperature | Chilled mirror hygrometer 1011B<br><br>-40 to +50 °C | Derived from the water-vapor mixing ratio, which is measured by a tunable diode laser (TDL) system (DLR development), 5-40000 ppmv |
| 3-D wind | Aircraft Integrated Meteorological Measurement System 20 (AIMMS-20) | Instrumented nose boom tray (DLR development) with an air data probe (Goodrich/Rosemount) 858AJ and high-precision Inertial Reference System (IGI IMU-IIe) |
| Particle number concentration | CPC, cut off size ($D_p$) =10 nm | CPC, cut off size ($D_p$) =10 nm |
| Size distribution* | UHSAS-A, 60-1000 nm. | UHSAS-A, 60-1000 nm. |
| | FIMS, 20 nm – 500 nm | |
| Non-Refractory particle chemical composition | HR-ToF-AMS: Organics, Sulfate, Nitrate, Ammonium, Chloride, 60-1000 nm | C-ToF-AMS: Organics, Sulfate, Nitrate, Ammonium, Chloride, 60-1000 nm |
| CCN concentration | CCN-200, SS= 0.25, 0.5% | CCN-200, SS= 0.13-0.53% |
| Gas phase concentration | N2O/CO and Ozone Analyzer, CO, $O_3$ concentration, precision 2 ppb | N2O/CO and Ozone Analyzer, CO, $O_3$ concentration, precision 2 ppb |
| Cloud properties* | CDP, 2-50 µm, $\Delta D_p$=1-2 µm | CCP-CDP, 2.5-46 mm, $\Delta D_p$=1-2 µm |
| | FCDP, 2-50 µm, $\Delta D_p$=1-2 µm | NIXE-CAS: 0.61 -52.5 µm |
| | 2DS, 10-1000 µm | NIXE-CIPgs, 15-960 µm |
| | | CCP-CIPgs: 15-960 µm |
| Radiation | SPN1 downward irradiance, 400-2700 nm | SMART Albedometer, downward spectral irradiance, 300-2200 nm |

***for an individual flight, the size range may vary.**

Table 2. Summary of the total data points compared between the G1 and HALO instruments.

| | SEP 9, 2014 | | SEP 21, 2014 | |
|---|---|---|---|---|
| | G1 | HALO | G1 | HALO |
| **Atmospheric parameters** | 2815 | 2815 | 7326 | 12065 |
| **Gas phase, CO** | N/A | N/A | 7326 | 12065 |
| **Gas phase, Ozone** | 2815 | 2815 | 7110 | 11766 |
| **CPC** | 2043 | 2043 | 8466 | 11646 |
| **UHSAS (FIMS)** | 2031 | 2031 | 5841 (9405) | 828 |
| **AMS** | N/A | N/A | 587 | 818 |
| **CCNc** | 663 | 531 | 7982 | 4546 |
| **G1: CDP(FCDP) HALO: CCP-CDP (NIXE-CAS)** | N/A | N/A | 3627(4439) | 2051(2260) |
| **G1: 2DS HALO: CCP-CIPgs (NIXE-CIPgs)** | N/A | N/A | 2280 | 2261 (2260) |
| **RAD** | 1355 | 1355 | N/A | N/A |

Table 3. Summary of basic statistics of data between in situ measurements

*Comparison of the coordinated flight on Sep. 9*

| *Variables* | G1 | | | | HALO | | | | | |
|---|---|---|---|---|---|---|---|---|---|---|
| | min | max | mean | std | min | max | mean | std | slope | $R^2$ |
| *T, K* | 297.7 | 300.2 | 298.9 | 0.5 | 297.2 | 299.4 | 298.4 | 0.4 | 1.002 | Neg. |
| *P, hPa* | 955 | 965 | 960.1 | 1.5 | 958 | 964.9 | 961.8 | 0.9 | 0.998 | Neg. |
| *WSpd, m/s* | 0.3 | 8.9 | 3.4 | 1.2 | 0.3 | 7.7 | 3.8 | 1.1 | 0.998 | Neg. |
| $T_{dew}$*, k* | 293 | 296.5 | 295.0 | 0.5 | 292.9 | 294.9 | 294.0 | 0.3 | 0.996 | Neg. |
| *Ozone, ppb* | 0.5 | 58.8 | 22.2 | 9.3 | 18.3 | 50.8 | 26.3 | 6.6 | 1.082 | 0.9401 |
| *CPC,* $cm^{-3}$ | 696.0 | 3480.6 | 1591.3 | 568.7 | 687.4 | 2639.4 | 1313.8 | 473.5 | 0.819 | 0.8508 |
| *UHSAS,* $cm^{-3}$ | 78.2 | 1118. | 645.5 | 116.3 | 504.1 | 1622.2 | 756.3 | 138.6 | 1.165 | 0.8193 |
| *CCNc (κ)* | 0.010 | 0.347 | 0.1855 | 0.067 | 0.012 | 0.394 | 0.1890 | 0.083 | 0.8937 | Neg. |

Table 4. List of compared measurements ranges and measurement variances caused by the spatial
variation during the field campaign.

| Measurement Variables | Measured Range during the Field Campaign | Measurement Variances between the Two Aircraft |
|---|---|---|
| Static Pressure | 500 – 1010 hPa | < 1 % |
| Static air temperature | 272 – 310 K | < 1% |
| Dewpoint temperature | 230 -300 K | Without clouds, <1%<br>With clouds, the measurement from the G1 can be up to 5% lower than that of HALO |
| 3-D wind | 1-15 m/s | < 40% |
| Particle number concentration | 500 – 15,000 $cm^{-3}$ | < 20% for CPC, <50% for UHSAS (size dependent) |
| Non-Refractory particle chemical composition | < 10 $\mu g \cdot m^{-3}$ | < 10% above 2500 m<br><br>Up to 50% below 2500 m |
| CCN concentration | SS=0.25%, 100 – 2000 $cm^{-3}$ | < 10% above 2500 m<br><br>Up to 50% below 2500 m |
| Gas phase concentration | Ozone: 15-75 ppb<br>CO: 50-200 ppb | Ozone: < 25%<br>CO: < 15% |
| Cloud droplet number concentration | 3- 20 $\mu m$ | <50 % |
| Downward irradiance | 200 -1500 $W \cdot m^{-2}$ | < 10% |

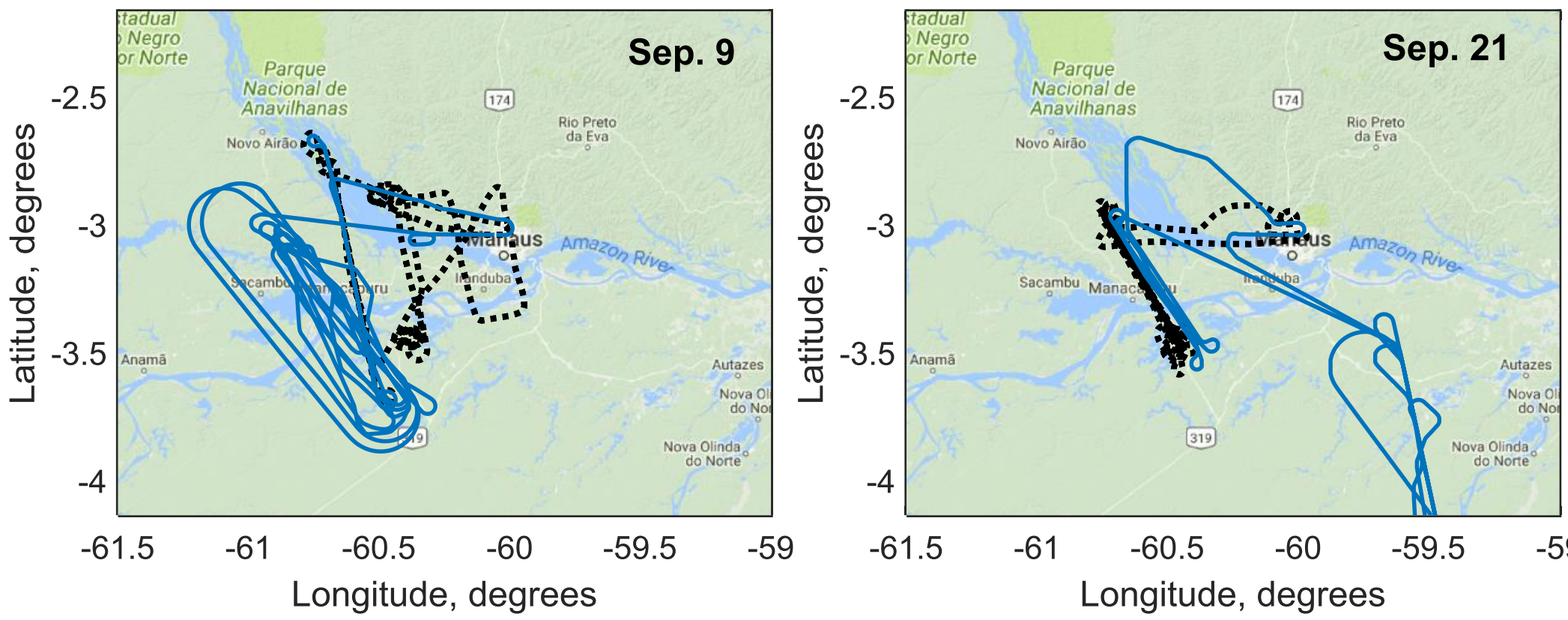

(a) (b)

Figure 1. Coordinated flight tracks for September 9 (a) and September 21 (b). The black dotted line is the flight track of the G1, and the blue line is the flight track of HALO.

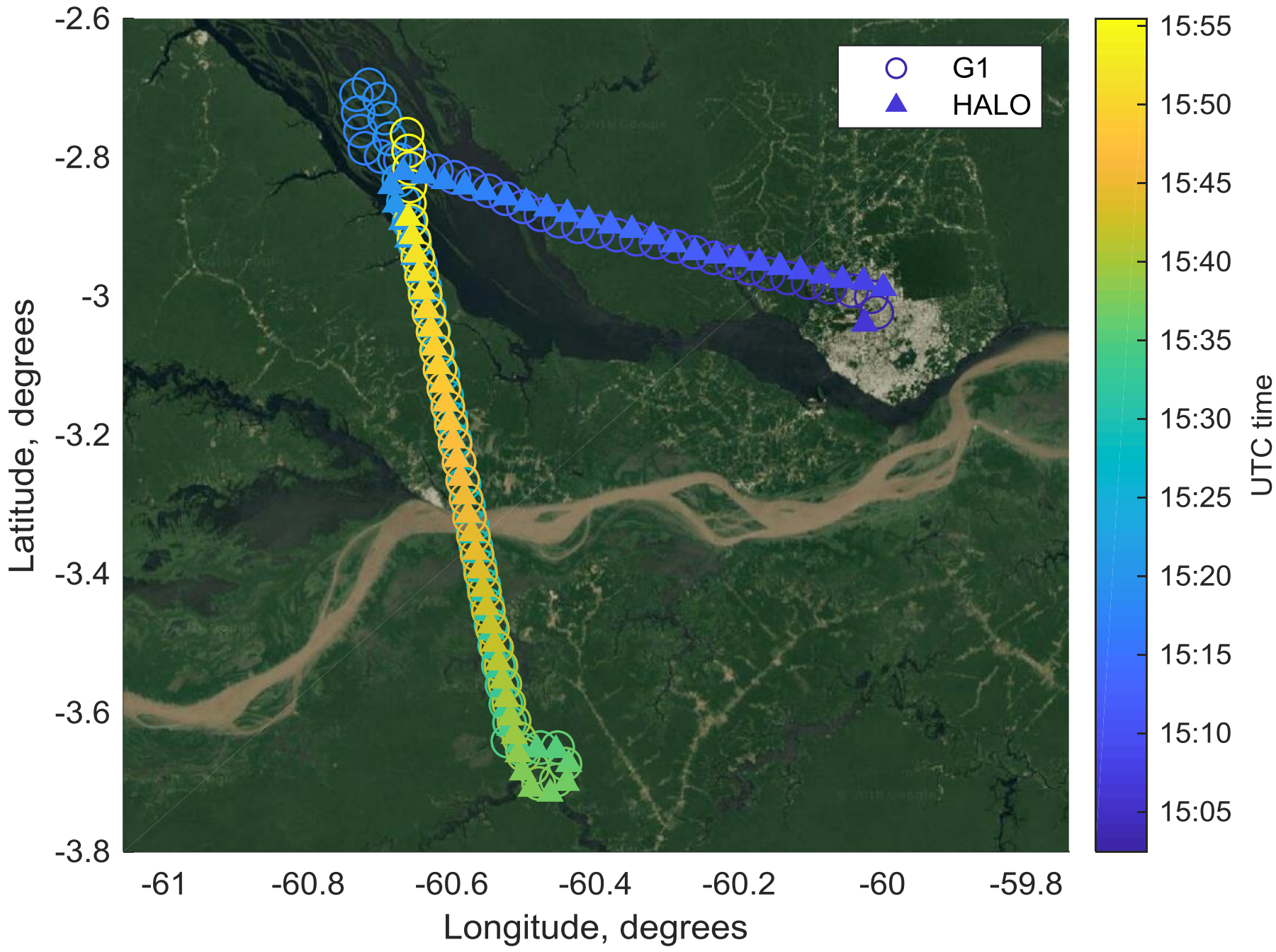

Figure 2. Time-colored flight track of the G1 (circle) and HALO (triangle) on September 9 during
a cloud-free coordinated flight at 500 m above sea level (50 m apart as the closest distance).

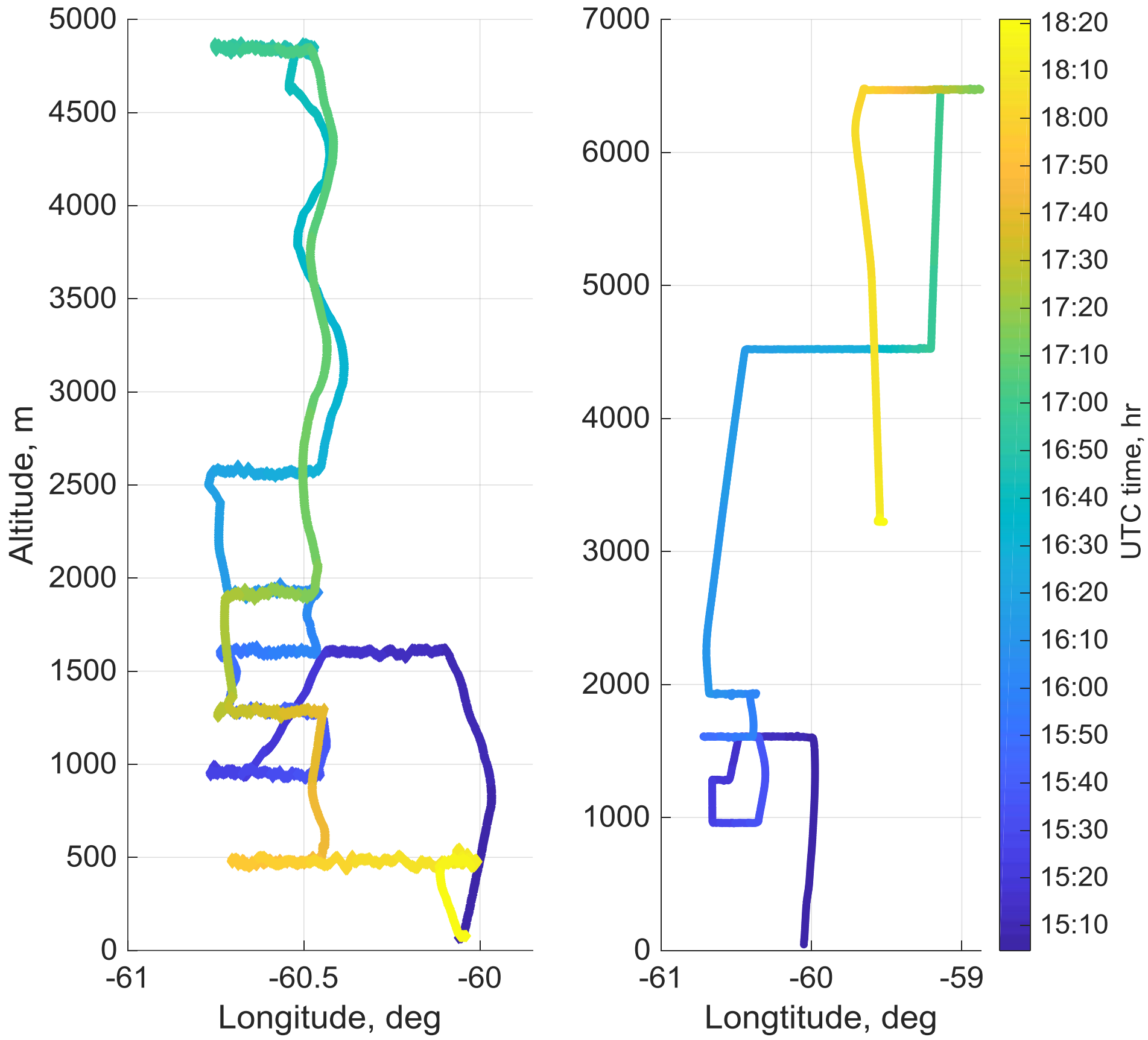

(a) (b)

Figure 3. Time-colored flight profile of the G1 (a) and HALO (b) on September 21, during a coordinated flight.

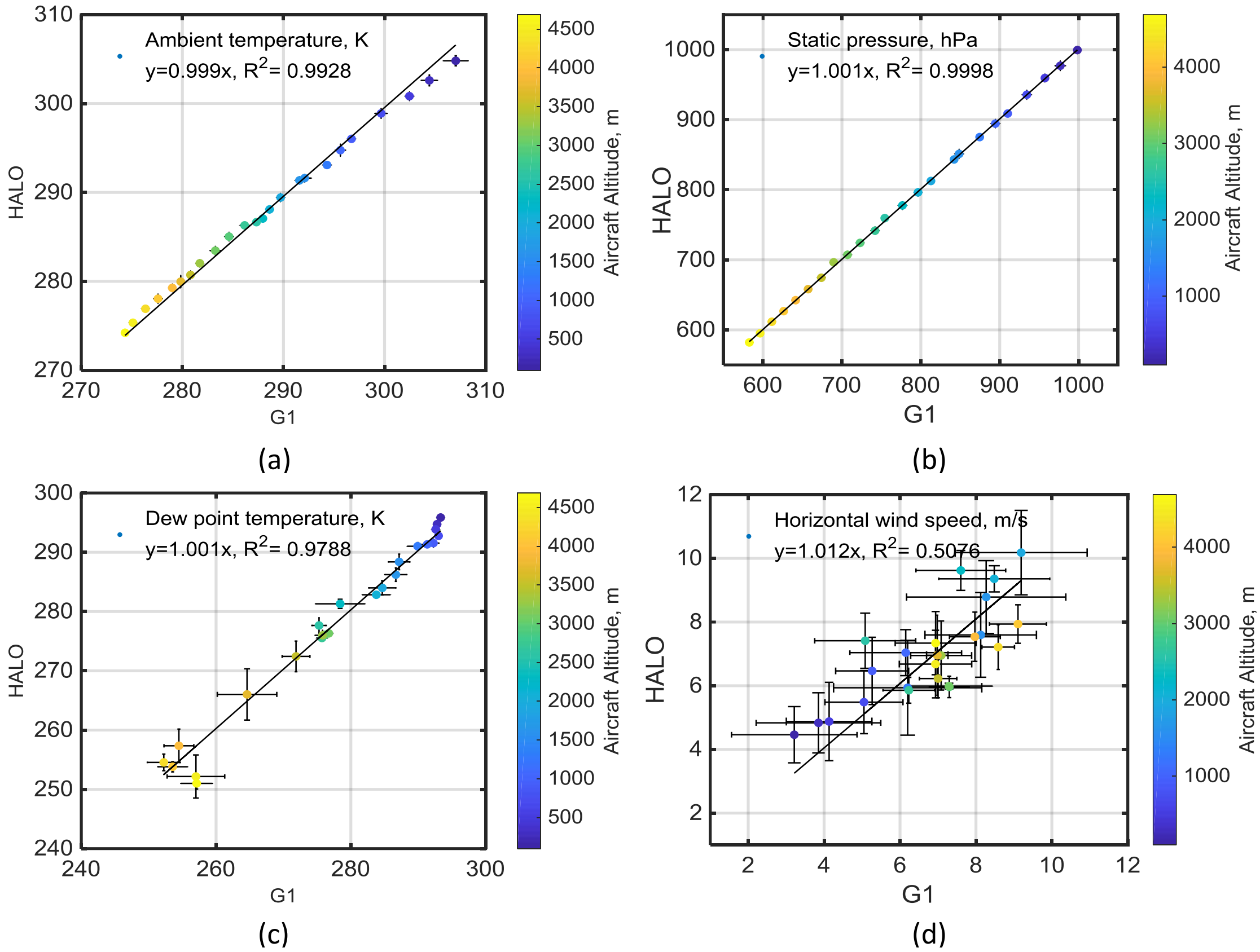

Figure 4. Aircraft altitude-colored plots of (a) ambient temperature, (b) static pressure, (c) dew point temperature, and (d) horizontal wind speed observed by the G1 and HALO on September 21.

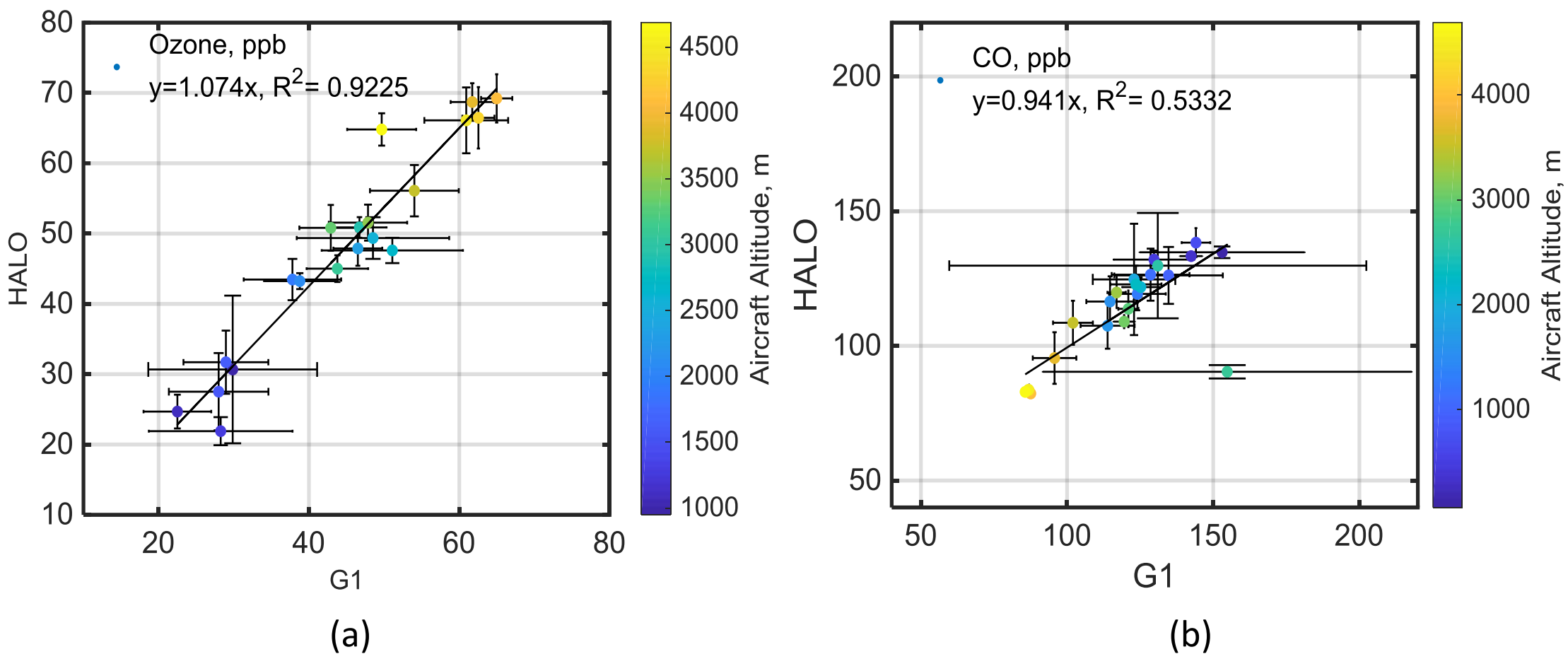

Figure 5. Aircraft altitude-colored plots of trace gas (a) Ozone, (b) CO, for the coordinated flight on September 21.

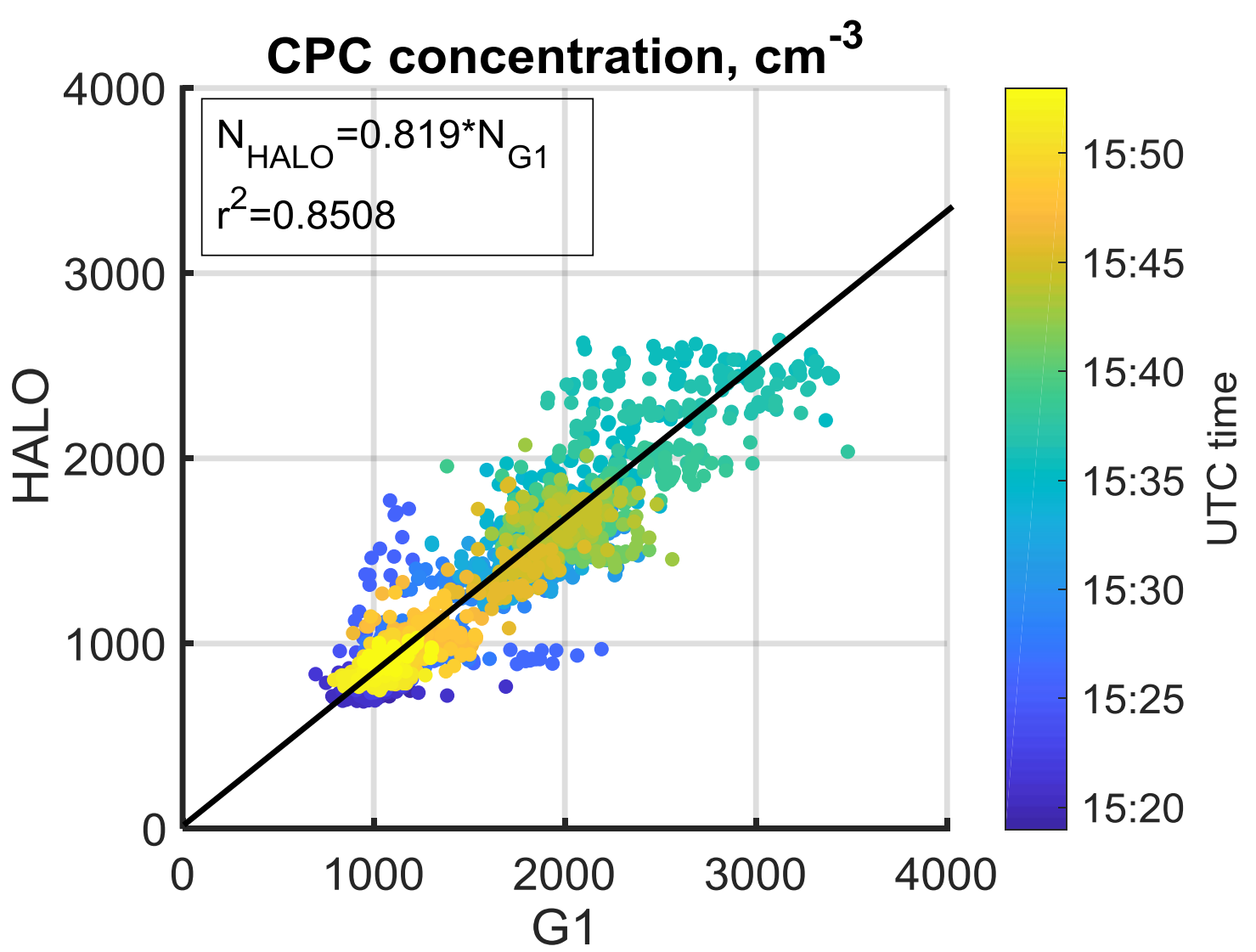

Figure 6. The G1 and HALO comparison for aerosol number concentration measured by CPC (>10 nm) on September 9.

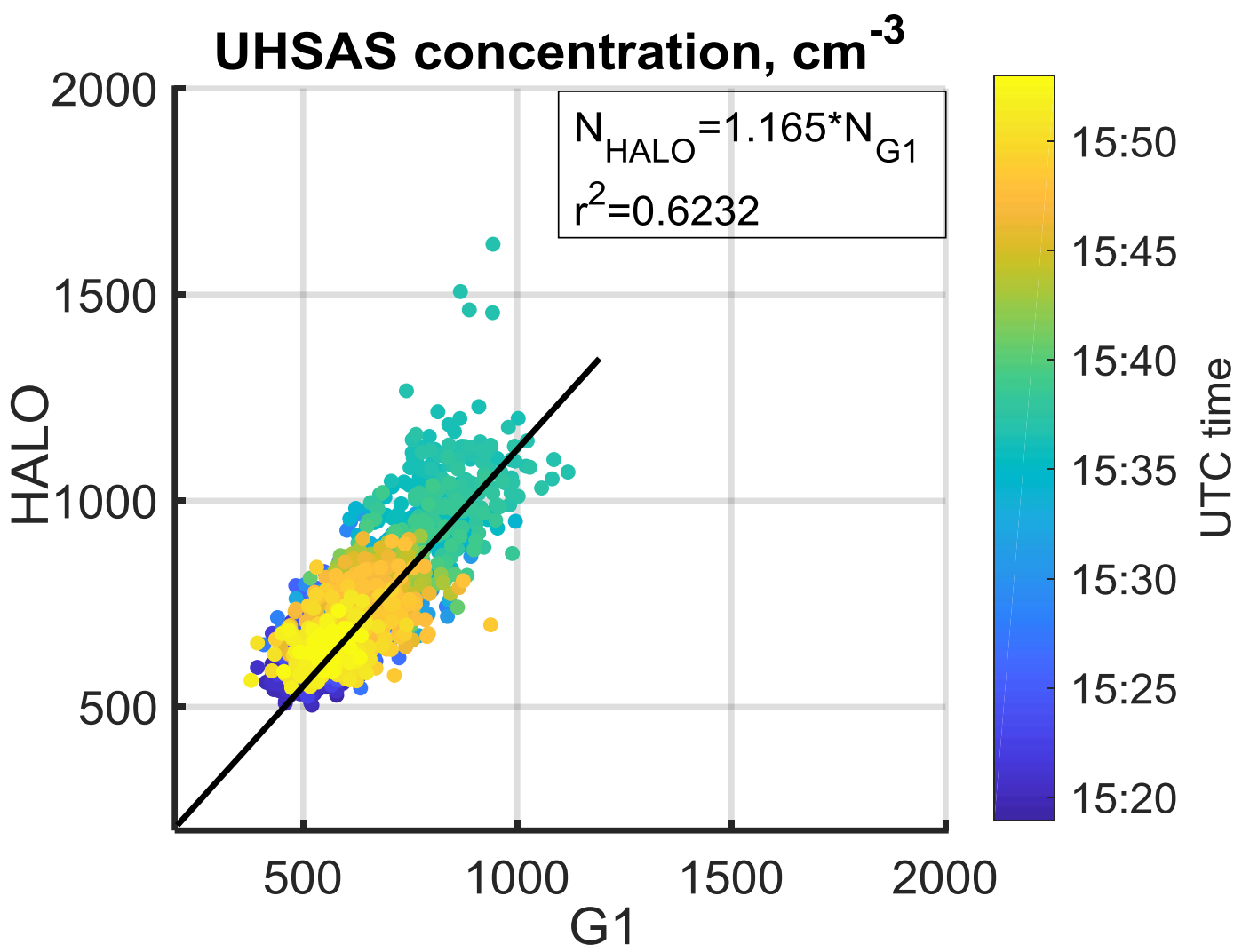

Figure 7. The G1 and HALO comparison for aerosol number concentration measured by UHSAS (90-500 nm) on September 9.

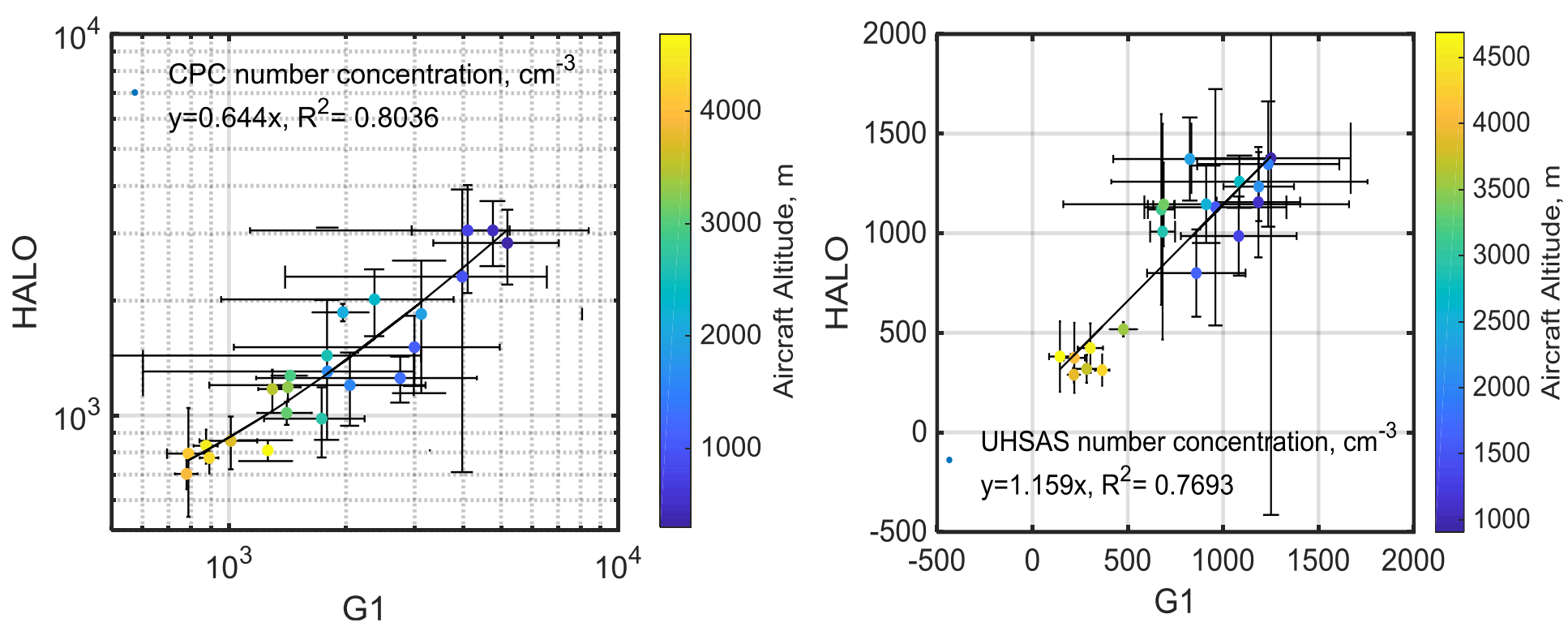

(a) (b)

Figure 8. The G1 and HALO comparison for aerosol number concentration profiling measured by (a) CPC and (b) UHSAS (100-700 nm) on September 21.

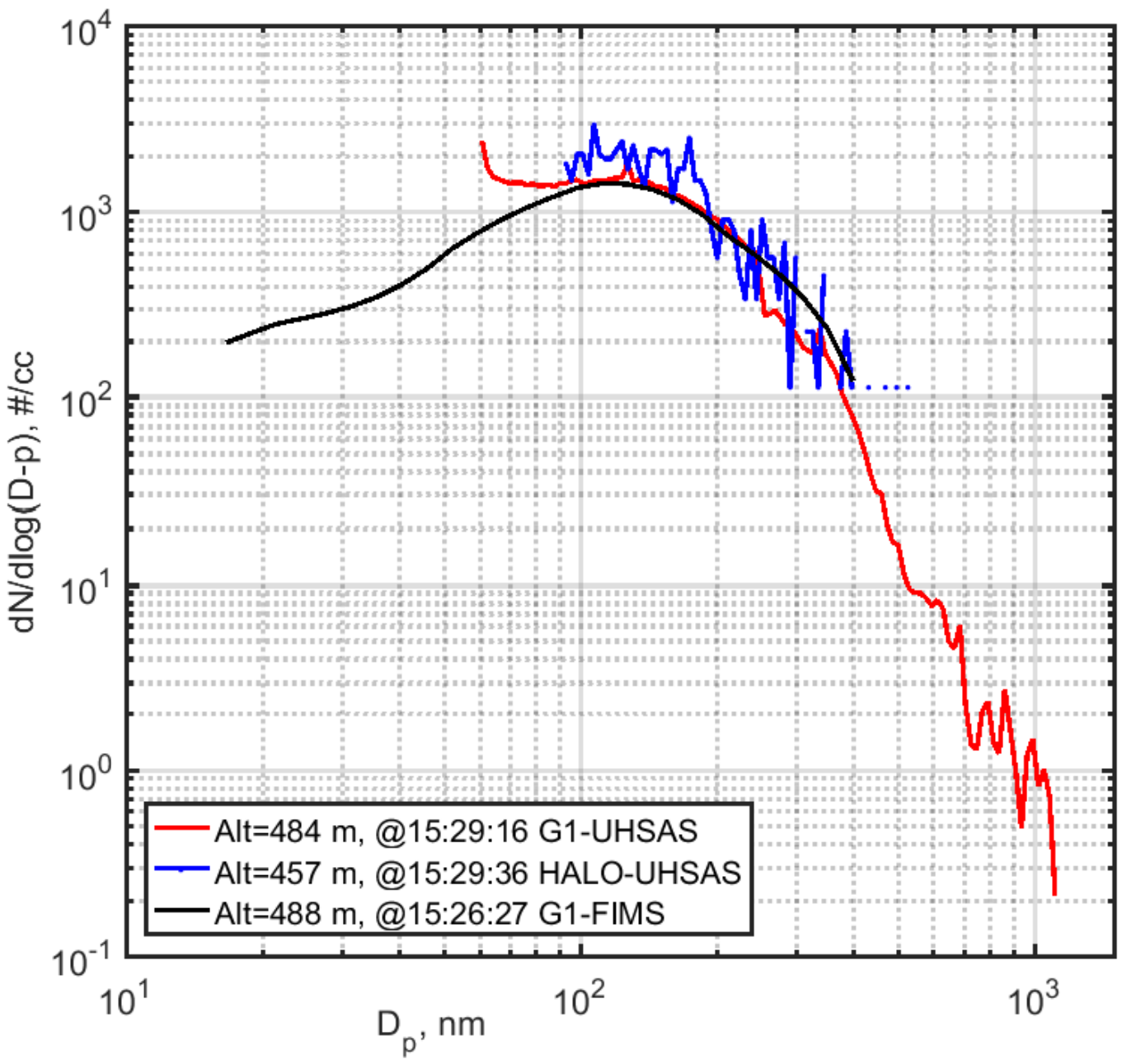

Figure 9. The G1 and HALO comparison for aerosol size distribution measured by UHSAS

(from both aircraft) and FIMS (on the G1) on September 9.

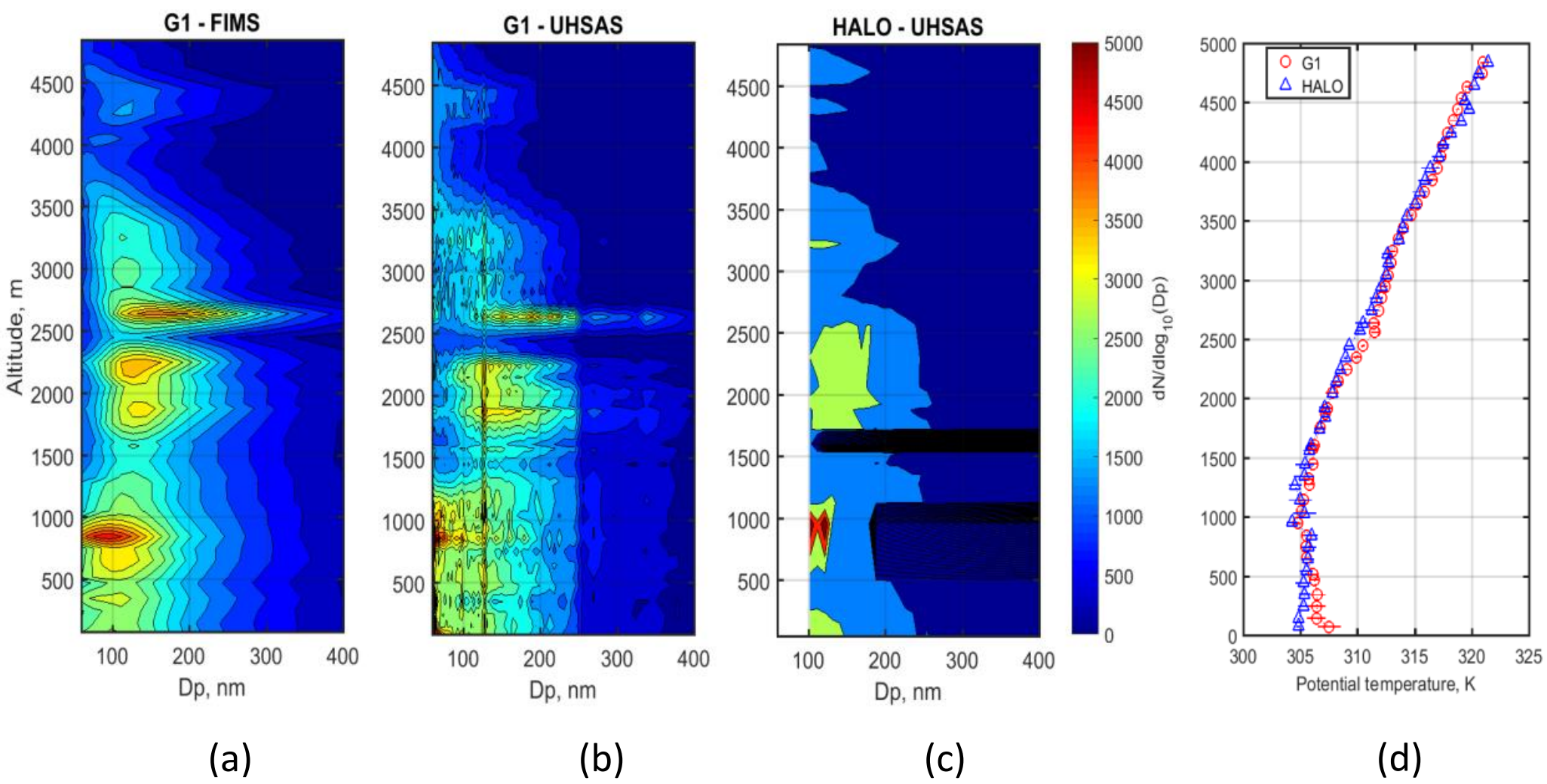

Figure 10. Aerosol size distribution vertical profiles measured by (a) the G1 FIMS, (b) The G1 UHSAS, (c) the HALO UHSAS, (d) Potential temperature aboard the G1 and HALO on September 21.

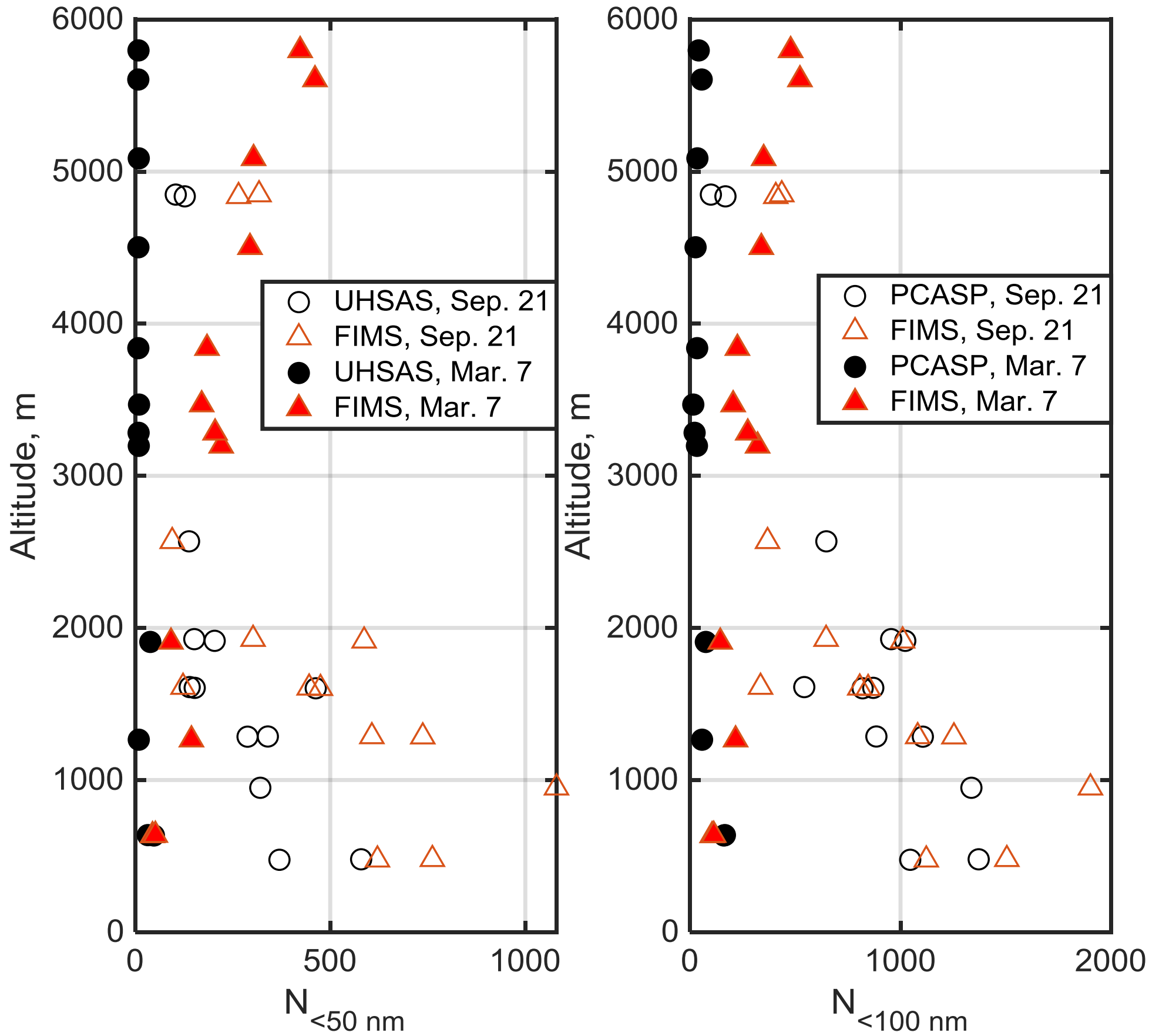

(a) (b)

Figure 11. Comparison of the number concentration for (a) sizes less than 50 nm and (b) sizes
less than 100 nm. FIMS-measured size distribution and UHSAS/PCASP estimated size
distribution are used.

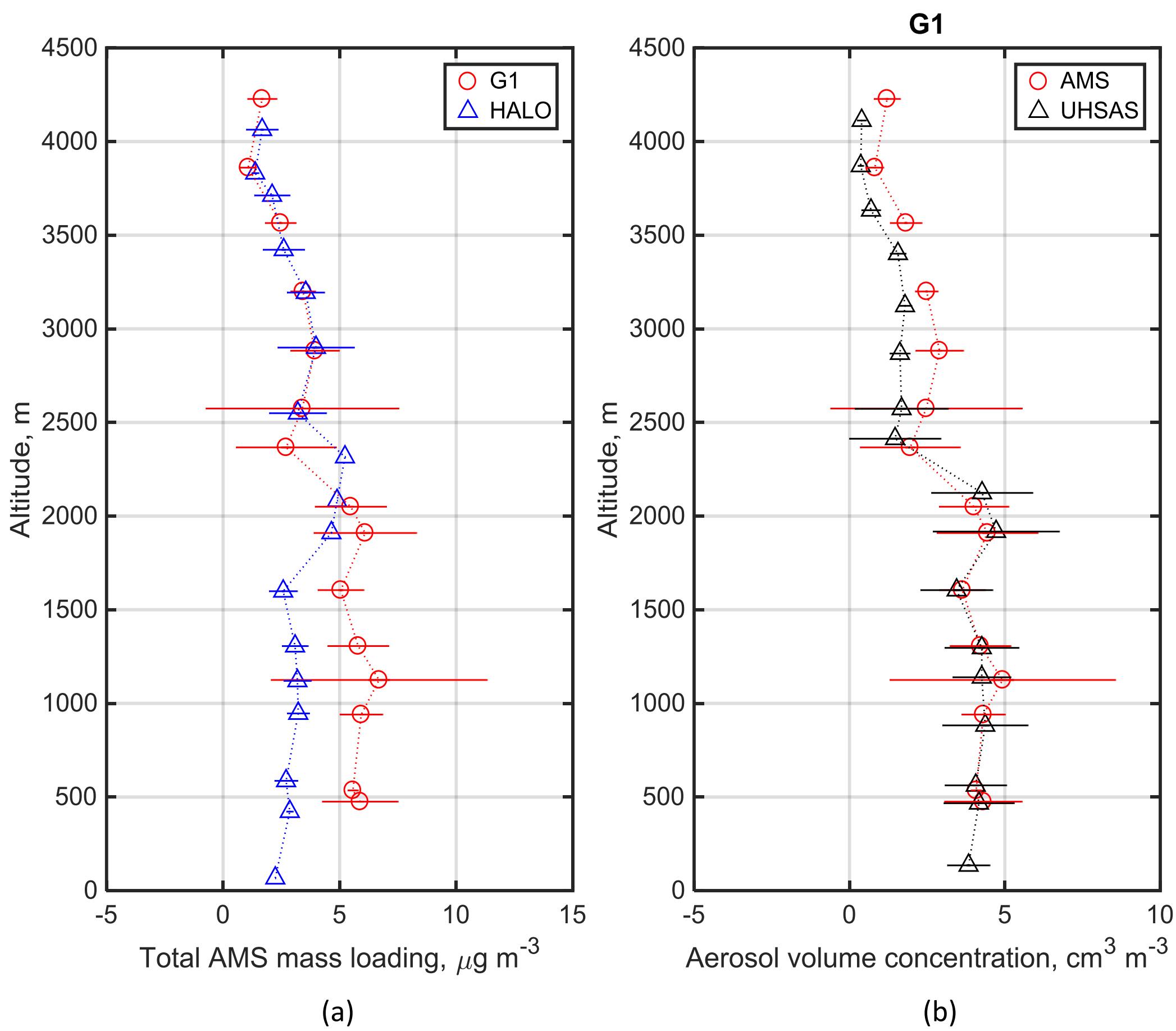

Figure 12. (a) Comparison of aerosol mass loading measured by the G1 and HALO AMS on September 21; (b) aerosol volume concentration comparison from AMS and the integrated UHSAS on the G1.

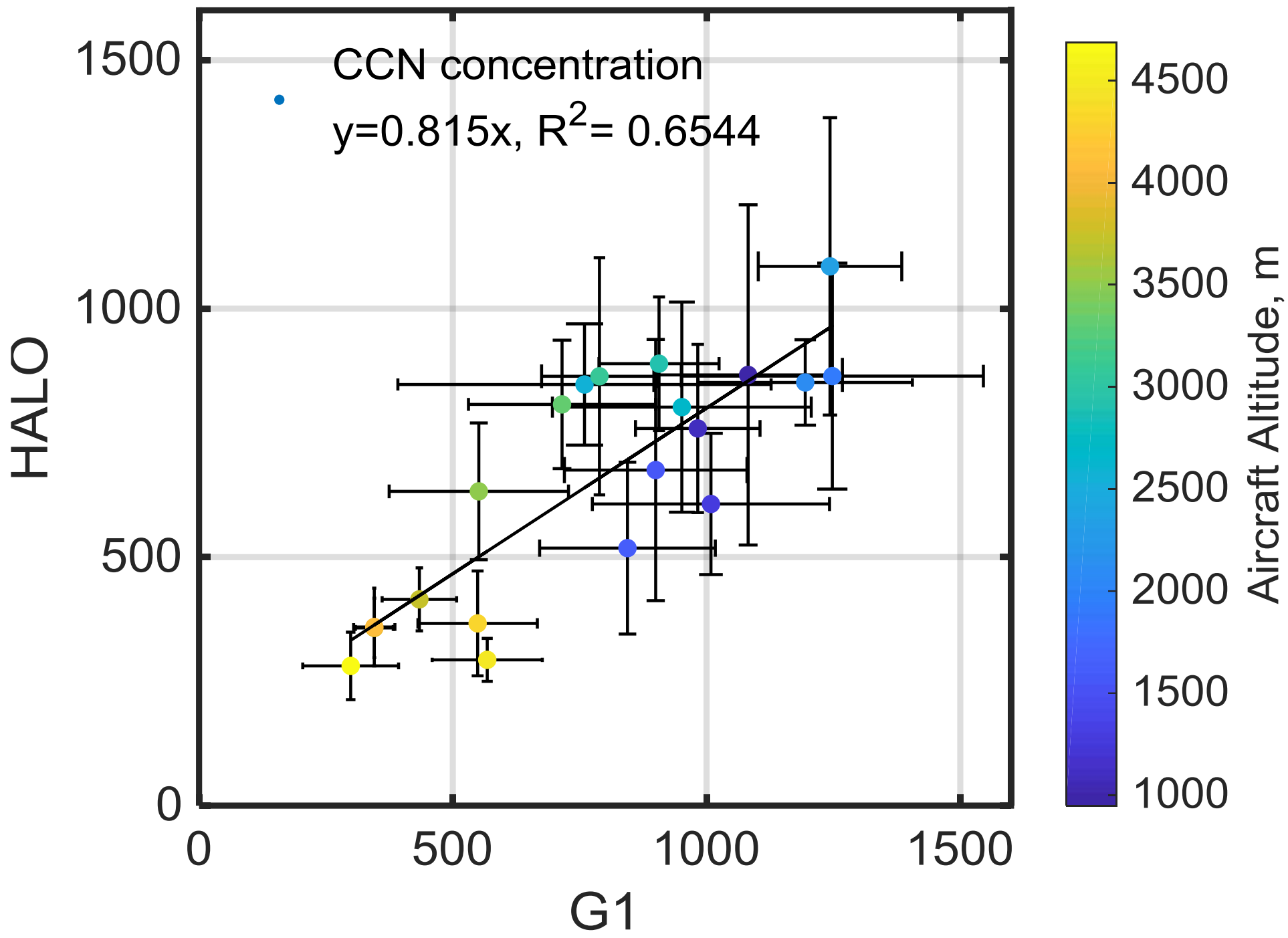

Figure 13. The G1 and HALO comparison of aerosol CCN concentration (S=0.5%) measured

on September 21.

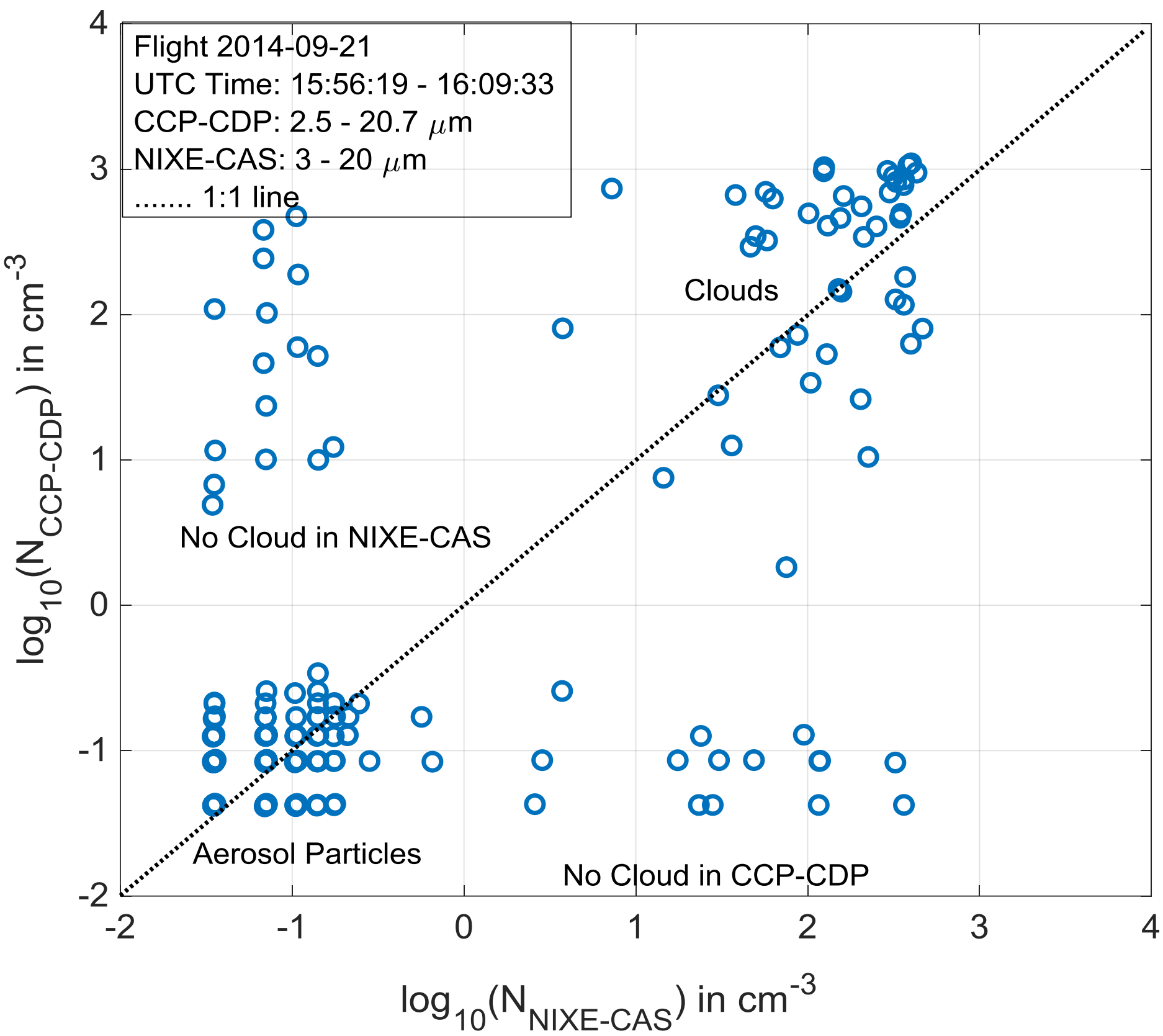

(a)

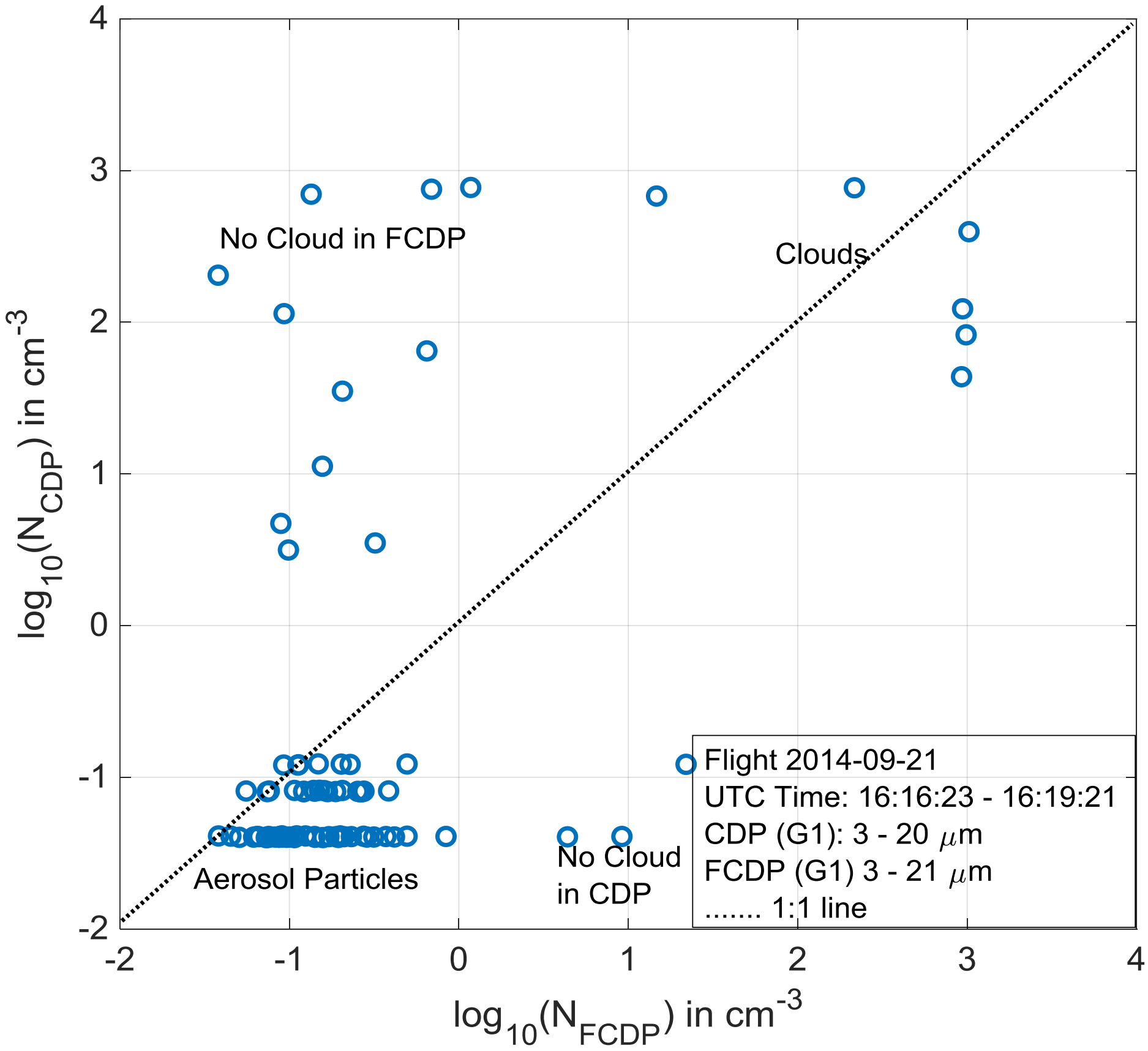

(b)

Figure 14 The comparison of cloud droplet concentrations in the same aircraft (a) between NIXE-CAS and CCP-CDP on board HALO; (b) between CDP and FCDP on board the G1.

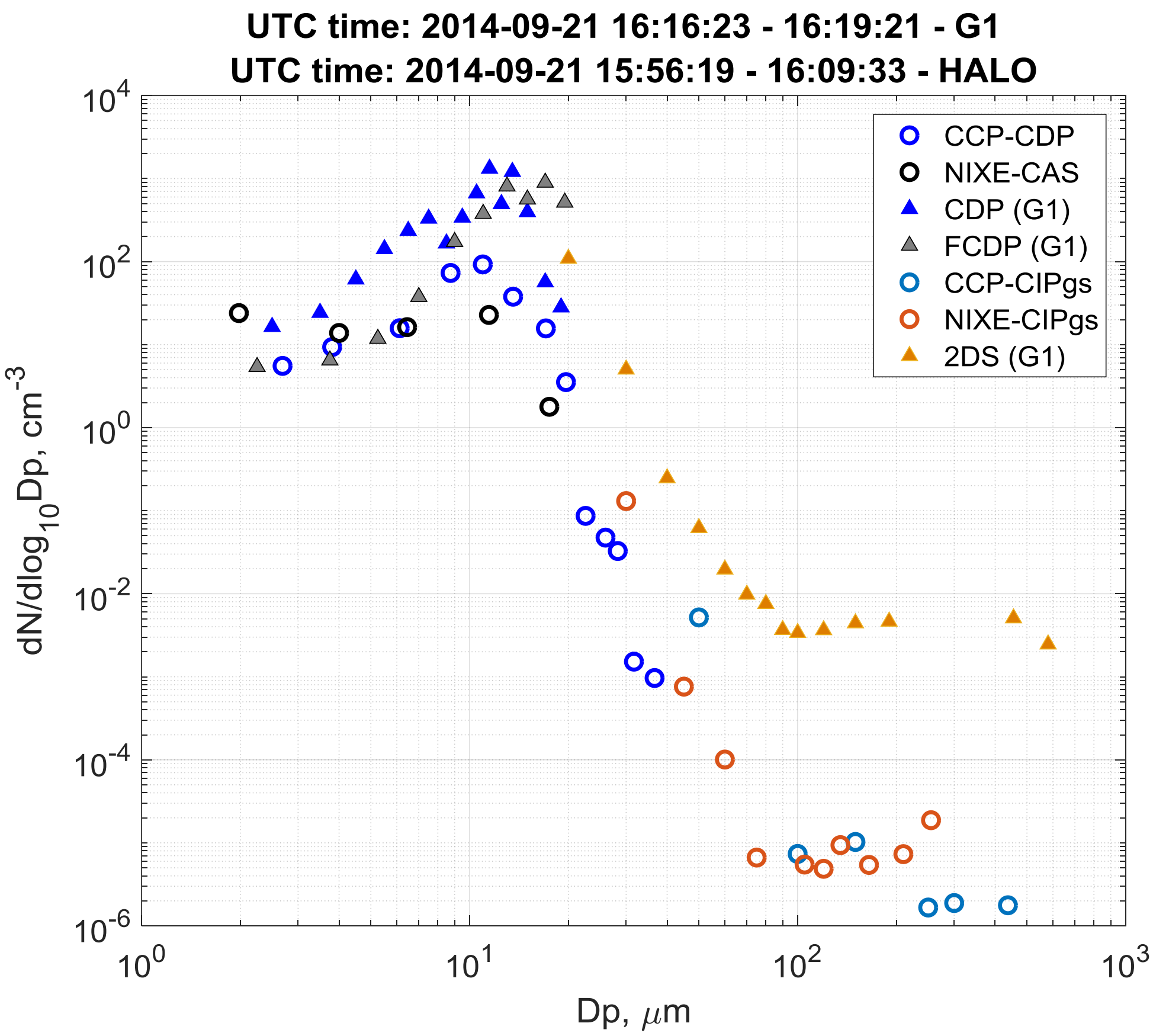

Figure 15. The cloud droplet size distribution from the cloud probes on the G1 and HALO.

1176

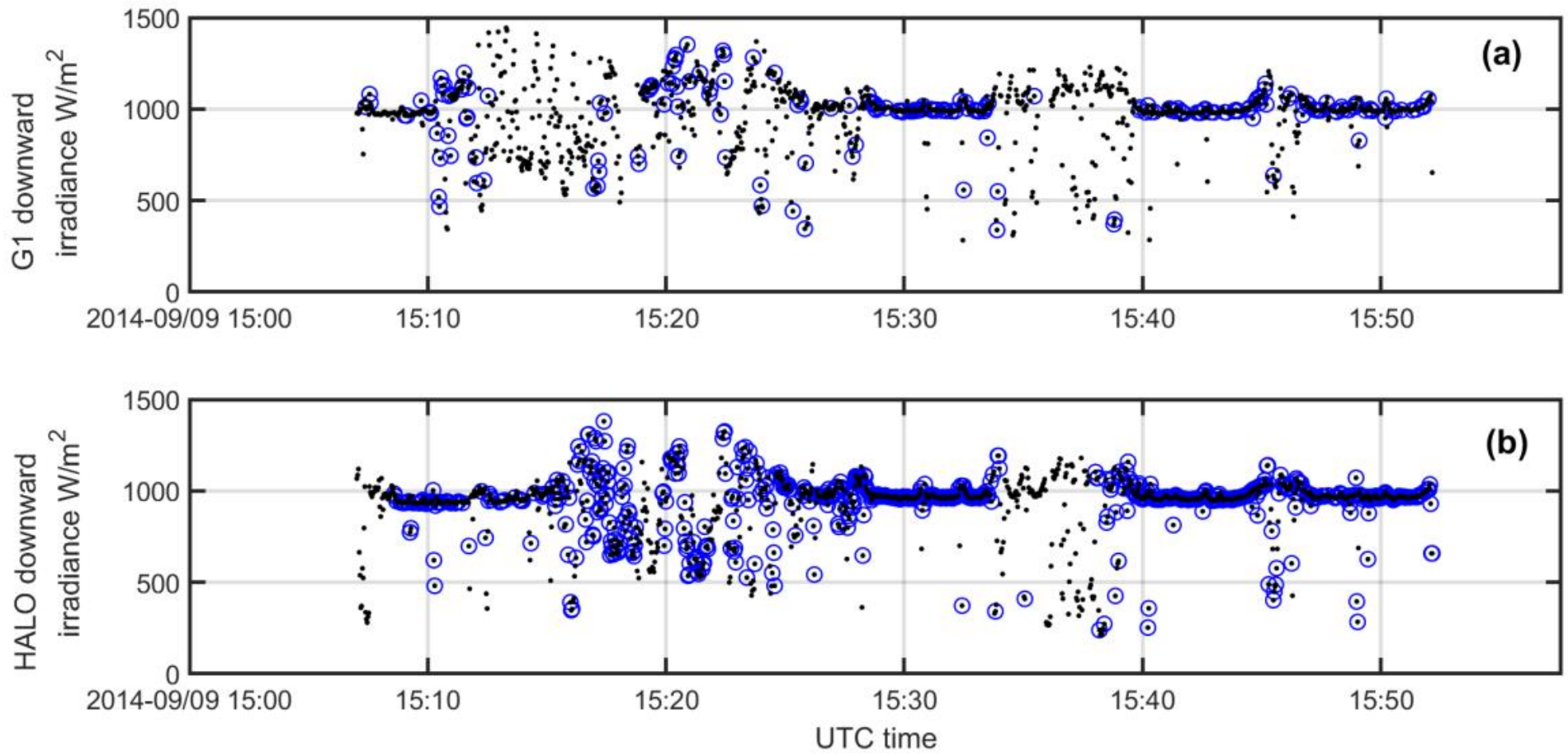

Figure 16. Time series of the G1 and HALO downward irradiance on September 9. The (a) by SPN-1 and (b) by SMART-Albedometer. Black dots represent all data under the general inter-comparison criteria. The blue circles represent the restricted navigation criteria.

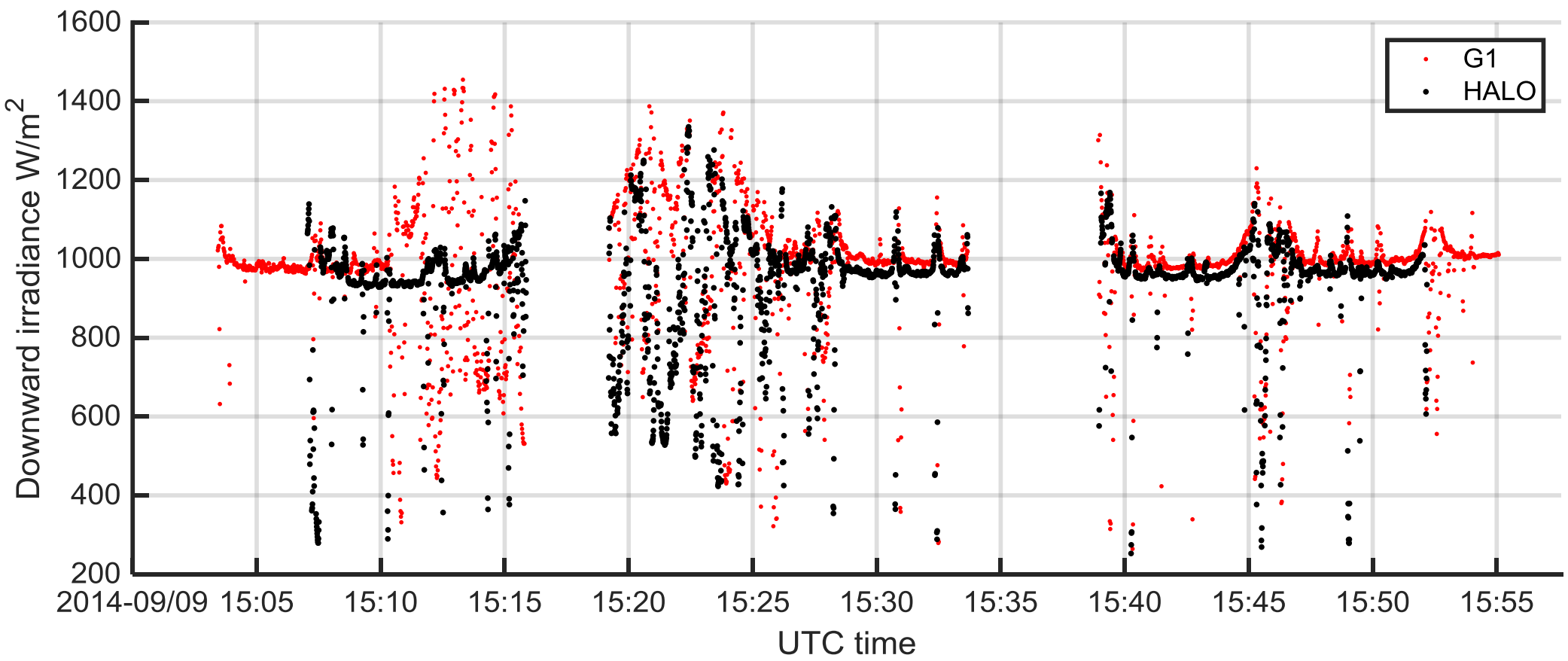

Figure 17. Time series comparison of the G1 (SPN-1) and HALO (SMART-Albedometer) radiation measurements on September 9.