# Peer review of "Comparison of Aircraft Measurements during GoAmazon2014/5 and ACRIDICON-CHUVA"

_Atmospheric Measurement Techniques, 2019_

## Referee Comment (RC1) · Anonymous Referee #1 · 24 May 2019

Review of "Comparison of aircraft measurements during GoAmazon2014/5 and ACRIDICON-CHUVA" by Mei et al.

Recommendation: Reject

This paper looks at data from two different aircraft (the G1 and HALO) collected during three coordinate flights during a recent research campaign, and reports on the comparisons between the measurements obtained by these aircraft. Although comparisons of the performance of instruments operating on the same platform are rather common, the intercomparison between probes operating on different aircraft are less common (but not non-existent). Therefore, it is possible that detailed comparisons between probes could yield some interesting findings as it would allow one to better assess how the different flight conditions as opposed to the instruments themselves affect the

intercomparison.

Unfortunately, this paper does not make a significant contribution to the literature in this respect. Instead, it merely presents data on what were the differences between instruments measuring a range of different variables (e.g., gas phase measurements, aerosols, clouds and solar radiation) and comparing these differences against the quoted uncertainties in the measurements. I really did not learn much by reading the manuscript as there really was not detailed treatment of why any of the differences between the instruments occurred, but rather only a very broad brush cursory description of what the differences were was included.

This paper read much like a technical document that many different groups who operate aircraft write after the completion of a field project. The question thus arises as to whether such technical documents should appear in the refereed literature as this group is trying to do. My thought is that such a paper does not belong in the refereed literature. If such papers are to be routinely published by all groups, there will be a massive proliferation of published papers. And, this paper really does not make a meaningful contribution to understanding the operating characteristics of different probes as would be expected from an AMT manuscript. Thus, whereas there are no flaws in the manuscript as submitted, I do not see that the manuscript makes any substantial contribution to scientific progress within the scope of the journal (substantial new concepts, ideas, methods or data). Therefore I recommend rejection.

If the authors would want to convert this work into a refereed publication, I would encourage them to do more detailed comparisons of the instruments in their different categories separately (e.g., gas phase, clouds, aerosols, radiation, etc.). This would allow them to delve into more details of how the probes actually work and examine whether or not the different aircraft parameters also have some effect on the measured quantities. Such a study would have more scientific significance.

---

## Referee Comment (RC2) · Anonymous Referee #4 · 6 Jul 2019

Experimental studies of the atmosphere using aircraft are extremely important and multi-aircraft experiments are often performed to expand the range of measurements or the spatial or temporal scales. Sometimes comparisons between these different aircraft platforms are performed and are often very instructive for those involved since they improve the measurements and identify any issues with the data processing or instruments. However, they are rarely published. It is therefore good to see that the authors are trying to provide this for the major study carried out above the Amazon region. These exercises are often very important and allow the data sets from both aircraft to be combined and integrated together. This is very useful and the paper achieves this aim by providing statistical comparisons between the platforms. In this sense it provides a useful contribution to the ACRIDICON-CHIVA experiment.

[Figure]

Unfortunately, it does little more and this is an opportunity missed. It would have been very good to see a more insightful discussion of the instrument performance, pre and post flight calibration details and what happens if these are not carried out. What, if any ground comparisons were carried out and how useful these were to the overall performance of the instruments? Were data analysis approaches compared and what did these yield? A more detailed discussion of these topics would provide some real insight and information for others carrying out similar work, whether in a single aircraft project or when multiple aircraft are being used together. I would strongly recommend that this is carried out in a revised manuscript and that some of the sections are removed such as the extrapolation of the size distribution to smaller sizes and the radiation sections.

Comments Line 193-207: Given that there are some discrepancies in the AMS measurements it would be very useful to have more information on the inlets and sample tubing for the two instruments, particularly the pressure controlled inlet systems. Were the instruments calibrated before and after each flight or if not when were the calibrations performed? Were all the instrument parameters (ionization efficiency/air beam, flow rates etc) varying in a consistent way throughout the experiment? How was the CE determined?

Line 414-420: No comment is made about the two sets of points at the start of the comparison which show enhancements in aerosol number in both aircraft at separate times, presumably one shortly after another. This gives rise to an increase in the uncertainty statistics but not the regression since the values are relatively low. It might also be good to discuss the breadth of points in the CPC regression since it could almost be argued that the pairs of points fall around two different regression lines.

Lines 434-439: If you can demonstrate that the aerosol sources are systematically different in the two profiles from the G1 and HALO then I don't see any justification for including the plot in the paper since there is no information to be gained. I suggest a clearer and more detailed explanation of why the aerosol sources in the two measured profiles are different and then a statement stating that this is the reason for not including the comparison, or if this cannot be satisfactorily demonstrated the statement of causality should be removed.

Line 471: I would recommend the removal of section 3.3.3. This is already a long paper and contains considerable amounts of detailed information. This section doesn't really show any comparison as such, it simply says that extrapolating a particle number size distribution below 100 nm based on optical particle size distribution information alone will underestimate the particle concentration if there is a small aerosol mode. In deep convection such particles can be activated and so extrapolations are to be treated with caution in environments where this occurs. A comment to this effect in the previous section is important as a caution but reducing the text would certainly help also.

Line 519 and following: Despite Section 3.3.4 being titled aerosol composition there is no comment about the chemical composition only a focus on the transmission of one of the AMS inlets. The implication from what is written is that the aerosol is predominately organic. Some discussion of the composition and any difference between the two instruments discussed. This is particularly the case if the inorganic components are above the detection limit since one could then test the effectiveness of the ion balance to derive ammonium concentrations. It would also be good to include some comment on the Collection Efficiency that is used and how this was calculated.

The quality of the English, particularly through the cloud section is rather poor. This needs to be significantly improved before publication.

Lines 618-619: It is always difficult to compare cloud probes between aircraft due to the spatial and temporal distances between the two aircraft. Nevertheless, this section does fall short of any detailed insight at all. It is stated that "The difference between the G1 CDP and FCDP may be due to the data post-processing". The implication is that this wasn't checked out in detail. There is clearly no information here that can be used by a reader that would be remotely useful. I suggest that much more detailed analysis is provided for this to be useful. Why weren't the corrections for coincidence

and shattering applied in a consistent manner?

Line 632 and following: This section says almost nothing at all and could be removed.

Line 652: Uncertainty Assessment: This section is extremely qualitative and non specific. As written it serves very little purpose. Instead I would recommend a much more detailed examination of uncertainties embedded with each of the sections and for this to be made more quantitative.

Minor comments: (I stopped writing the minor corrections after a while since the latter part of the paper needs a significant revamp if it is to remain).

Line 85: uncertainty ranges

Line 101: issues

Line 1100: delta

Line 155: section 2.1.3: were the CPCs from the G1 and the HALO run side by side on the ground for a period? If so it would be good to report this. When were the instruments calibrated relative to the field experiment? This isn't said explicitly.

Line 188-189: which have a refractive index

Line 178-192: when were the UHSAS instruments calibrated relative to the flight periods?

Line 227: should read in present tense "are discussed"

Line 234: needs to be rewritten "working independently and electronics produce shadowgraph"

Line 263: not sure about the use of the word "proven"

Lines 247-268: How were the sample volumes of the HALO probes established? This is stated for the G1 but not HALO.

Lines 276-278: It is not clear how this is actually achieved.

Line 304: stacked pattern

Lines 308-309: "Due to the different aircraft speeds, the flight 309 distance between two aircraft flight paths continued increasing from 15 min to 1 hour" I do not dispute that the distances between the flight paths continued increasing but since the G1 took off first and the HALO is the faster aircraft I cannot see how the increase in time between the aircraft is due to the different aircraft speeds.

Line 321: present

Line 323: intervals

Line 337: "The linear regression achieved a slope was near 1" should be "The linear regression achieved a slope that was near 1"

Line 340-342: This is a good way of presenting the uncertainty though I am surprised that you didn't use the orthogonal distance that would also represent the variability in x.

Line 356: when (the) G1 flew

Line 418: change "rest" for "remaining"

Line 387-393: I am unsure why the regression statistics are presented including the point with high CO measured by the G1 but not by the HALO in Fig 5b. By all means present the data point but it does seem a little strange to include it in the reporting of the agreement.

Line 439: sources

Line 470: "has a reduced spatial resolution"

Line 718: strophic?

---

## Author Comment (AC1) · 15 Jul 2019

We appreciate the referee efforts in reviewing our manuscript. However, we regret that the referee does not consider the manuscript worthy of publication. For reasons that we explain in the following, we do not share this opinion. 1. Many of the authors of this manuscript have participated in twenty plus aircraft measurement campaigns. Based on their experiences, a merely technical document comparing instruments to each other for each campaign is not a common practice. 2. Thus, we feel that such comparison studies should appear in the referenced literature to educate the community on how to examine the measurements obtained from more than one aircraft, including pointing out potential issues. This valuable information, is very important to share with a broader audience. 3. Also, publications of instrument intercomparisons

from field or laboratory campaigns are common in the refereed literature, in AMT and other journals. Here are a few examples:

- Brock et al. (2019): "Aerosol size distributions during the Atmospheric Tomography Mission (ATom): methods, uncertainties, and data products", https://www.atmos-meas-tech.net/12/3081/2019/amt-12-3081-2019-discussion.html

- Meyer et al. (2015): Two decades of water vapo measurements with the FISH fluorescence hygrometer: A review., ACP, 15, 8521–8538, https://doi.org/10.5194/acp-15-8521-2015, 2015.

- Fahey et al (2014): The AquaVIT-1 intercomparison of atmospheric water vapor measurement techniques, AMT, 7, 3177–3213, https://doi.org/doi:10.5194/amtd-7-3177-2014.

- Rollins et al. (2014): Evaluation of UT/LS hygrometer accuracy by intercomparison during the NASA MACPEX mission, JGR, 119, 1915–1935, https://doi.org/doi:10.1002/2013JD020817, 2014.
* * *

---

## Referee Comment (RC3) · Anonymous Referee #5 · 26 Aug 2019

Mei et al. provide a comparison of datasets from two research aircraft obtained during a coordinated comparison effort. Comparisons between calibrated instruments are quite useful for evaluating whether the estimated uncertainties for each instrument do accurately represent the data quality, which is of course paramount to the usefulness of the data. Ideally, analysis of such comparisons could be used to better understand estimated uncertainties and possibly reduce those uncertainties.

This paper takes on a significant effort because the authors compare all of the possible measured parameters between these two aircraft (> 10 parameters). In general, I feel that the paper would be more useful if the scope were somewhat smaller with more significant analysis and discussion of the differences between a subset of the measurements. There are a few useful recommendations for measurements going

forward, but also some of the disagreements between measurements which might be considered significant are not explored enough to understand if the measurements can be reconciled. At the same time, I don't think it is reasonable to ask the authors to change what they see as the purpose of the paper, but suggest that in the future such comparisons may better serve the community by going more in-depth on a smaller group of the measurements.

I have some suggestions and changes that I would like to see the authors address. These are listed below.

Table 3: Recommend instead of highlighting only slope and R2 that the systematic differences in measurements are calculated (two measurements could be perfectly correlated with a slope of 1 yet have a huge offset and differ on average by a large fraction). Can you also include something about the expected agreement based on the uncertainties of each instrument?

Line 365: Was the fact that the G1 sensor data point bad here known before the comparison and would it have been thrown out? If so, recommend removing this point from the figure as it does represent what is thought to be good data.

Section 3.2: Ozone: Table 3 shows a minimum ozone value of 0.5 for G1. Is this correct or a typo? It seems there is a slope and offset between the ozone instruments. Difference between the means is about 17%, which I think exceeds what is expected ($\sim$ 5% each instrument). I doubt the explanation that sampling losses in the tubing could account for the difference as O3 is not too difficult to sample. Please state clearly whether the differences observed between the O3 instruments exceeds what is expected for the sensors themselves, and what evidence there is to suggest sampling loss is to blame. Possibly, a leak of cabin air into the sample line affected one of the instruments.

CO: Recommend removing the outlier CO point if you have good reason to believe it was not coincident. At the same time, I don't see how the explanation on 389-391 about

"different operation principles" has anything to do with lack of coincidence between the measurements. Please clarify if the disagreement is because of bad coincidence or if you think the instruments really do not measure the same thing.

Line 418: Kind of weak discussion here about CPC difference. Seems like HALO is systematically lower. It would be useful to understand something about the difference rather than just state that it can be attributed to the typical uncertainties and other unknown factors. The comparison between UHSAS does not support it being an issue with the isokinetic inlets.

Figure 9: Why does HALO UHSAS look so much noisier?

Line 488 / Fig. 11: I don't see the value of this comparison. It is stated in the text that the UHSAS < 50 nm is not a measurement, but rather an extrapolation of the distribution down to sizes the UHSAS cannot measure, and that this extrapolation could easily be invalid during e.g. a nucleation event. Therefore, I don't understand when the extrapolated UHSAS data would ever be of use for scientific analysis. The fact that the extrapolated UHSAS distribution deviates from the FIMS measurements sometimes does not even require the UHSAS instrument to determine this. One could just extrapolate the FIMS data using the UHSAS sensitivity range and look at the difference between the FIMS measurements.

Line 512: What is referred to here had been done for decades on other aircraft and has been referred to as NMASS. Recommend citing the relevant papers for that here and earlier in the paper (i.e. lines 478 – 487). Most recently: Williamson et al., AMT 11, 3491-3509, 2018.

Line 519/section 3.3.4: There is no actual discussion of the chemical composition, just the mass/volume. Recommend removing the reference to chemical composition here and earlier in the paper (e.g. abstract and introduction).

L 650: How about a calculation with TUV to test whether the different sensitivity ranges

can account for the 10%?

————- Editorial type notes:

Line 101: issures -> issues

Line 146: change comma to period

Line 304: 'paten' -> 'pattern'?

Line 326: 'Tables' -> 'Table'

Line 418: ' rest of the 10-15..."

---

## Author Comment (AC2) · 20 Sep 2019

**RC2**: 'referee report', Anonymous Referee #4

Experimental studies of the atmosphere using aircraft are extremely important and multi-aircraft experiments are often performed to expand the range of measurements or the spatial or temporal scales. Sometimes comparisons between these different aircraft platforms are performed and are often very instructive for those involved since they improve the measurements and identify any issues with the data processing or instruments. However, they are rarely published. It is therefore good to see that the authors are trying to provide this for the major study carried out above the Amazon region. These exercises are often very important and allow the data sets from both aircraft to be combined and integrated together. This is very useful and the paper achieves this aim by providing statistical comparisons between the platforms. In this sense it provides a useful contribution to the ACRIDICON-CHIVA experiment.

Response: We thank the reviewer for all the valuable comments and appreciate the suggestions the reviewer made. Below, we have added our responses to the comments submitted.

Unfortunately, it does little more and this is an opportunity missed. It would have been very good to see a more insightful discussion of the instrument performance, pre and post flight calibration details and what happens if these are not carried out. What, if any ground comparisons were carried out and how useful these were to the overall performance of the instruments? Were data analysis approaches compared and what did these yield? A more detailed discussion of these topics would provide some real insight and information for others carrying out similar work, whether in a single aircraft project or when multiple aircraft are being used together. I would strongly recommend that this is carried out in a revised manuscript and that some of the sections are removed such as the extrapolation of the size distribution to smaller sizes and the radiation sections.

Response: Thank you very much for your suggestions. We changed the manuscript to provide more insights about the instruments' performance: 1) we included more details in the aerosol chemical composition comparison section 3.3.3 and added two figures (Fig 11 and 12); 2) we expanded the discussion in the trace gas and aerosol number concentration section 3.2 ; 3) we modified cloud probe comparison section to include details on the data processing (Line 587-596

and Line 628-636); 4) we ameliorated the comparison results with an additional linear regression approach (Table 4). More details are provided in the responses to specific comments.

Comments Line 193-207: Given that there are some discrepancies in the AMS measurements it would be very useful to have more information on the inlets and sample tubing for the two instruments, particularly the pressure controlled inlet systems. Were the instruments calibrated before and after each flight or if not when were the calibrations performed? Were all the instrument parameters (ionization efficiency/air beam, flow rates etc) varying in a consistent way throughout the experiment? How was the CE determined?

Response: The constant pressure inlets used by both G1 and HALO AMS were very similar to the design by Bahreini et al. 2008 (the reference had been included in section 2.1.3). Both AMS instruments were calibrated before and after the field deployment and also once a week during the field campaign (line 210-211). More details about the AMS measurements are given in separate AMS papers from the respective groups (Schulz et al., 2018; Shilling et al., 2018). A short summary is now included in the supplemental material. For example, the CE of the G1 AMS was determined by comparing AMS data to UHSAS and FIMS data. We confirmed the CE=0.5 by comparing mass loadings observed at the T3 site to the G1 data. For HALO AMS, CE of 0.5 was applied, as recommended by Middlebrook et al. (2012) for low nitrate conditions.

Line 414-420: No comment is made about the two sets of points at the start of the comparison which show enhancements in aerosol number in both aircraft at separate times, presumably one shortly after another. This gives rise to an increase in the uncertainty statistics but not the regression since the values are relatively low. It might also be good to discuss the breadth of points in the CPC regression since it could almost be argued that the pairs of points fall around two different regression lines.

Response: We included further discussion of the CPC difference in lines 443 – 457 with additional plots in Figure 6.

Lines 434-439: If you can demonstrate that the aerosol sources are systematically different in the two profiles from the G1 and HALO then I don't see any justification for including the plot in the

paper since there is no information to be gained. I suggest a clearer and more detailed explanation of why the aerosol sources in the two measured profiles are different and then a statement stating that this is the reason for not including the comparison, or if this cannot be satisfactorily demonstrated the statement of causality should be removed.

Response: Based on the aerosol number concentration, chemical composition, and CO concentration data, we believe that the G1 and HALO were sampling different air masses at altitudes between 2000 and 3000 m. Thus, we excluded these data points on replotted figure 8 and revised the corresponding discussion in section 3.3.1.

Line 471: I would recommend the removal of section 3.3.3. This is already a long paper and contains considerable amounts of detailed information. This section doesn't really show any comparison as such, it simply says that extrapolating a particle number size distribution below 100 nm based on optical particle size distribution information alone will underestimate the particle concentration if there is a small aerosol mode. In deep convection such particles can be activated and so extrapolations are to be treated with caution in environments where this occurs. A comment to this effect in the previous section is important as a caution but reducing the text would certainly help also.

Response: We moved this section to the supplemental material. The original objective was to emphasize the importance of expanding size distribution measurements below 50 nm range on an airborne platform using advanced instrumentation (e.g., FIMS). We learned from many modelers that they typically extrapolate UHSAS size distributions in their models due to scarcity of actual data below ~50 nm.  Thus, we compared the measurement from FIMS to the UHSAS based extrapolations.

Line 519 and following: Despite Section 3.3.4 being titled aerosol composition there is no comment about the chemical composition only a focus on the transmission of one of the AMS inlets. The implication from what is written is that the aerosol is predominately organic. Some discussion of the composition and any difference between the two instruments discussed. This is particularly the case if the inorganic components are above the detection limit since one could

then test the effectiveness of the ion balance to derive ammonium concentrations. It would also be good to include some comment on the Collection Efficiency that is used and how this was calculated.

Response: More discussion of the chemical composition was added to the manuscript in section 3.3.3 on page 19-20.

The quality of the English, particularly through the cloud section is rather poor. This needs to be significantly improved before publication.

The revised manuscript has been reviewed by many native speaking co-authors and the cloud section is edited by a professional editor.

Lines 618-619: It is always difficult to compare cloud probes between aircraft due to the spatial and temporal distances between the two aircraft. Nevertheless, this section does fall short of any detailed insight at all. It is stated that "The difference between the G1 CDP and FCDP may be due to the data post-processing". The implication is that this wasn't checked out in detail. There is clearly no information here that can be used by a reader that would be remotely useful. I suggest that much more detailed analysis is provided for this to be useful. Why weren't the corrections for coincidence and shattering applied in a consistent manner?

Response: Thank you very much for your suggestions. We have modified the text accordingly. "The difference between the G1 CDP and FCDP is mainly due to the data post-processing. The G1 CDP used an earlier version of the data acquisition system from Science Engineering Associates, with limited capability to store the particle-by-particle (PBP) data for further processing. The CDP had an 800-$\mu$m-diameter pinhole placed in front of the sizing detector to minimize the coincidence up to 1850 cm$^{-3}$. On the other hand, FCDP was equipped with new electronics and PBP data was locally stored on a flash drive onboard the Linux machine. For the G1 flights, a constant probe-dependent adjustment factor was applied to FCDP to adjust the coincidence further. The G1 CDP and FCDP operated with redesigned probe tips to minimize the shattering effect. An additional algorithm was applied to the FCDP data to eliminate particles with short interarrival times."

Line 632 and following: This section says almost nothing at all and could be removed.

Response: This radiation comparison section is mainly to illustrate the challenges of comparing two radiation instruments deployed on two aircraft, including many factors which affect the accuracy of the measurements. And we also confirmed the effects of the difference in spectral sensitivity of the radiometers using the NCAR tropospheric ultraviolet and visible (TUV) radiation model.

Line 652: Uncertainty Assessment: This section is extremely qualitative and non specific. As written it serves very little purpose. Instead I would recommend a much more detailed examination of uncertainties embedded with each of the sections and for this to be made more quantitative.

Response: We have modified the uncertainty assessment section contents (page 23, section 4) and revised Table 4 to be quantitative. The information about the sources for the discrepancy between the two measurements can be useful for users to understand data uncertainty and for future field campaign planning.

Minor comments: (I stopped writing the minor corrections after a while since the latter part of the paper needs a significant revamp if it is to remain).

Line 85: uncertainty ranges

Response: corrected

Line 101: issues

Response: corrected

Line 1100: delta

Response: corrected

Line 155: section 2.1.3: were the CPCs from the G1 and the HALO run side by side on the ground for a period? If so it would be good to report this. When were the instruments calibrated relative to the field experiment? This isn't said explicitly.

Response: Unfortunately, we never got a chance to run CPCs from the G1 and HALO side by side on the ground. The aircraft were parked far apart, and such a comparison would have required un-mounting and relocation of one CPC, which was not practical during tightly scheduled field campaign. All CPCs were calibrated before and after the field campaign and checked at least once a week during the deployment.

Line 188-189: which have a refractive index

Response: corrected

Line 178-192: when were the UHSAS instruments calibrated relative to the flight periods?

Response: The UHSAS was calibrated before and after the field deployment and checked with PSL's once a week during the deployment.

Line 227: should read in present tense "are discussed"

Response: corrected

Line 234: needs to be rewritten "working independently and electronics produce shadowgraph"

Response: changed to "working independently. The 2DS electronics produce shadowgraph…"

Line 263: not sure about the use of the word "proven"

Response: changed to "examined".

Lines 247-268: How were the sample volumes of the HALO probes established? This is stated for the G1 but not HALO.

Response: line 257-258, "The sample area of the CCP- CDP was determined to be $0.27\pm0.025$ mm$^2$ with an uncertainty of less than 10% (Klingebiel et al., 2015)."

Lines 276-278: It is not clear how this is actually achieved.

Response: Dr. Long have provided more details about the tilt correction in his paper (Long et la., 2010). We modified the sentence to make it clear, "Additionally, the angular offset between the

actual orientation of each radiometer's detector and the level position from the navigation data has also been determined and corrected after the installation for each deployment."

Line 304: stacked pattern

Response: corrected.

Lines 308-309: "Due to the different aircraft speeds, the flight distance between two aircraft flight paths continued increasing from 15 min to 1 hour" I do not dispute that the distances between the flight paths continued increasing but since the G1 took off first and the HALO is the faster aircraft I cannot see how the increase in time between the aircraft is due to the different aircraft speeds.

Response:  We have revised the sentence: "Due to the different aircraft speeds, the time difference between two aircraft visiting the same part of the flight path varied, increasing up to 1 hour at the end of the path, as shown in Figure 3."...

Line 321: present

Response: corrected.

Line 323: intervals

Response: corrected

Line 337: "The linear regression achieved a slope was near 1" should be "The linear regression achieved a slope that was near 1"

Response: corrected

Line 340-342: This is a good way of presenting the uncertainty though I am surprised that you didn't use the orthogonal distance that would also represent the variability in x.

Response: Thank you very much for your good suggestions. We have included the orthogonal regression in new Table 4.

Line 356: when (the) G1 flew

Response: corrected

Line 418: change "rest" for "remaining"

Response: changed to "rest of the 10-15%"

Line 387-393: I am unsure why the regression statistics are presented including the point with high CO measured by the G1 but not by the HALO in Fig 5b. By all means present the data point but it does seem a little strange to include it in the reporting of the agreement.

Response: We replotted the CO comparison and modified the discussion in lines 397-419.

Line 439: sources

Response: corrected.

Line 470: "has a reduced spatial resolution"

Response: corrected.

Line 718: strophic?

Response: changed to "lower troposphere"

---

## Author Comment (AC3) · 20 Sep 2019

Mei et al. provide a comparison of datasets from two research aircraft obtained during a coordinated comparison effort. Comparisons between calibrated instruments are quite useful for evaluating whether the estimated uncertainties for each instrument do accurately represent the data quality, which is of course paramount to the usefulness of the data. Ideally, analysis of such comparisons could be used to better understand estimated uncertainties and possibly reduce those uncertainties. This paper takes on a significant effort because the authors compare all of the possible measured parameters between these two aircraft (> 10 parameters). In general, I feel that the paper would be more useful if the scope were somewhat smaller with more significant analysis and discussion of the differences between a subset of the measurements. There are a few useful recommendations for measurements going forward, but also some of the disagreements between measurements which might be considered significant are not explored enough to understand if the measurements can be reconciled. At the same time, I don't think it is reasonable to ask the authors to change what they see as the purpose of the paper, but suggest that in the future such comparisons may better serve the community by going more in-depth on a smaller group of the measurements.

We thank the reviewer for the thoughtful comments and suggestions. We also agree that the community may benefit from in-depth comparison of a smaller group of the measurements in the future. Our responses to the specific comments are described below.

I have some suggestions and changes that I would like to see the authors address. These are listed below.

Table 3: Recommend instead of highlighting only slope and R2 that the systematic differences in measurements are calculated (two measurements could be perfectly correlated with a slope of 1 yet have a huge offset and differ on average by a large fraction). Can you also include something about the expected agreement based on the uncertainties of each instrument?

Response: The fitting slope and R2 in Table 3 assume the correlation between two measurements: the G1 measurement is equal to HALO measurement. Thus, the listed slope is from y=slope* x equation without the offset, and the R2 is based on the equation 1. We modified the line 336 "assuming that two measurements from the G1 and HALO have the 1:1 relationship." We also tried the orthogonal regression to relate the measurements from the G1 and HALO, which shows the similar results as Table 3. We added another table (new Table 4) to further illustrate the uncertainties of each instrument.

Line 365: Was the fact that the G1 sensor data point bad here known before the comparison and would it have been thrown out? If so, recommend removing this point from the figure as it does represent what is thought to be good data.

Response: We replotted Figure 4(c) without the bad data points, then changed the sentence (line 366). The initial data quality control did not exclude the G1 sensor data as questionable data i.e. chilled mirror sensor wetted by cloud droplets.

Section 3.2: Ozone: Table 3 shows a minimum ozone value of 0.5 for G1. Is this correct or a typo? It seems there is a slope and offset between the ozone instruments. Difference between the means is about 17%, which I think exceeds what is expected ($\sim$ 5% each instrument). I doubt the explanation that sampling losses in the tubing could account for the difference as O3 is not too difficult to sample. Please state clearly whether the differences observed between the O3 instruments exceeds what is expected for the sensors themselves, and what evidence there is to suggest sampling loss is to blame. Possibly, a leak of cabin air into the sample line affected one of the instruments.

Response: We agree that the sampling loss is not the main reason causing the measurements difference. We have edited the lines 386-390 "As mentioned in section 2.1.2, each instrument has a 2 ppb accuracy (or 5%) on the ground based on a direct photometric measurement measuring the ratio between a sample and ozone free cell. The in-flight calibration showed that the variation of each instrument could raise to 5-7% (or 2-3.5 ppb). Thus, the difference between the averaged ozone concentrations – 4.1 ppb is close to the instrument uncertainty."

CO: Recommend removing the outlier CO point if you have good reason to believe it was not coincident. At the same time, I don't see how the explanation on 389-391 about "different operation principles" has anything to do with lack of coincidence between the measurements. Please clarify if the disagreement is because of bad coincidence or if you think the instruments really do not measure the same thing.

Response: We removed the outlier CO points from the altitude between 2000-3000 m, which we believe the G1 and HALO are sampling different air mass. We agree that the "different operation principles" should not cause the significant difference in the measurements. We modified the manuscript lines 397-419.

Line 418: Kind of weak discussion here about CPC difference. Seems like HALO is systematically lower. It would be useful to understand something about the difference rather than just state that it can be attributed to the typical uncertainties and other unknown factors. The comparison between UHSAS does not support it being an issue with the isokinetic inlets.

Response: We included further discussion of the CPC difference in between line 443 – 457 with additional figures in Figure 6.

Response: The airborne version of the UHSAS does have an issue with maintaining constant volumetric sheath flows (discussed by Kupc et al. 2017), which directly affects concentration since the sample flow is not directly measured but calculated as the difference between total and sheath flows.

Line 488 / Fig. 11: I don't see the value of this comparison. It is stated in the text that the UHSAS < 50 nm is not a measurement, but rather an extrapolation of the distribution down to sizes the UHSAS cannot measure, and that this extrapolation could easily be invalid during e.g. a nucleation event. Therefore, I don't understand when the extrapolated UHSAS data would ever be of use for scientific analysis. The fact that the extrapolated UHSAS distribution deviates from the FIMS measurements sometimes does not even require the UHSAS instrument to determine this. One could just extrapolate the FIMS data using the UHSAS sensitivity range and look at the difference between the FIMS measurements.

Response: We moved this section to the supplemental material. The original objective was to emphasize the importance of expanding size distribution measurements to below the 50 nm range on an airborne platform using advanced instrumentation (e. g. FIMS). We learned from many modelers that they typically use extrapolations of the UHSAS size distribution in their models due to scarcity of real data. Thus, we compared the real measurement from FIMS to the UHSAS based extrapolation.

Line 512: What is referred to here had been done for decades on other aircraft and has been referred to as NMASS. Recommend citing the relevant papers for that here and earlier in the paper (i.e. lines 478 – 487). Most recently: Williamson et al., AMT 11, 3491-3509, 2018.

Response: Thank you very much for your suggestions. The section is modified in the supplemental material.

Line 519/section 3.3.4: There is no actual discussion of the chemical composition, just the mass/volume. Recommend removing the reference to chemical composition here and earlier in the paper (e.g. abstract and introduction).

Response: More discussion of the chemical composition was added to the manuscript in section 3.3.3, pages 19-20.

L 650: How about a calculation with TUV to test whether the different sensitivity ranges can account for the 10%? ———-

Response: We used the tropospheric ultraviolet and visible (TUV) radiation model from NCAR website (https://www2.acom.ucar.edu/modeling/tropospheric-ultraviolet-and-visible-tuv-radiation-model) and estimated the weighted irradiance at 15:42:00 on Sep 9 2014. Note that the modeling output is limited to the range between 315 to 900 nm. It is different from the irradiance spectral range (400-2700 nm) in the G1 or the 300-1800 nm from HALO. The difference between the two aircraft measurements was 24.1 W/m^2 at that time, and the modeling suggested the irradiance difference between 315-400 nm was 13 W/m^2. Although we can't estimate the difference between 1800-2700 nm with TUV. We have shown that the difference in spectral range of the instruments is the main contribution to the difference in the comparison.

Editorial type notes: Line 101: issures -> issues

Response: corrected.

Line 146: change comma to period

Response: corrected.

Line 304: 'paten' -> 'pattern'?

Response: corrected.

Line 326: 'Tables' -> 'Table'

Response: corrected.

Line 418: ' rest of the 10-15. . ."

Response: corrected.

---

## Author Response (AR2)

Associate Editor Decision: Publish subject to minor revisions (review by editor) (22 Nov 2019)
by Wiebke Frey
Comments to the Author:
Dear Fan Mei et al.,
thanks for the revision of your manuscript.

As you know from the reviews the point of scientific significance of your manuscript has been controversely discussed. Since comparisons between performances of different instruments on different platforms (in this case the two research aircraft) are valuable for understanding benefits and limitations of the observations being made, thus being of interest to the scientific community, I decided to follow the suggestions of the reviewers that are in favour of publication of your manuscript.
However, please address the remaining reviewer comments (see below).

Best regards,
Wiebke Frey

**Response: We sincerely appreciate the support from our associate editor Wiebke Frey. Thank you very much for considering the publication of our manuscript. We address the reviewer's comments below (also in blue).**

Reviewer comments:
I commend the authors of this paper on making a number of improvements to the revised version of this paper. I really do think that comparisons such as this should be published and am pleased that the revised version of this paper has been resubmitted. I have rather few comments on the revision since the authors have done a good job of addressing my earlier concerns.

**Response: We thank the review for all the constructive comments and suggestions, which led to improvements in our manuscript. Our response to the comments is detailed below.**

The only area that I feel has not been addressed is in answer to my original comment "What, if any ground comparisons were carried out and how useful these were to the overall performance of the instruments? Were data analysis approaches compared and what did these yield?"
**Response: If any ground comparisons were carried out, it would help us further understand the sources of some of the differences observed. For example, some of the observed differences during the inter-comparison flights likely arise from the differences in aerosol inlet structures, aerosol particle losses correction in main aircraft inlets and/or constant pressure inlets (Line 457-458, Line 520-522, and line 570-572). A ground comparison would**

**help better identify the sources and quantify their contributions. Regarding the data analysis, standard procedures were followed for the same instruments on the two aircraft.**

I feel that though the authors now tell us what comparisons and calibrations were done they don't tell us how useful they were at improving the overall performance. How variable were the calibrations from day to day? What happens if a calibration was not performed on a particular day could average calibrations be used or not? What is the degradation of performance? This is helpful information and some discussion of this would be useful.

**Response: We thank the reviewer for these comments. Two tables are now added in the supplement, and one additional sentence in the main manuscript (Line 120-121) "Details on maintenance and calibration of the involved instrumentation can be found in the supplement (Table S2 and Table S3)." Due to the logistic difficulty, instruments mounted on the two aircraft discussed in this paper were not compared side-by-side on the ground. Calibrations and checks during the campaign were carried out on instrument flow rates and sizing, and they show minimal variation and can not explain the difference observed during the comparison flights. To provide more helpful information to the community, we include our recommendation to the end of the summary section.**

**"Several efforts made by both airborne measurement teams have significantly contributed to the overall success of this comparison study, and we recommend them for the future field operation.**

1) **Characterize instruments following the same established guideline. For example, the aerosol instruments can follow the guideline from the World Calibration Centre for Aerosol Physics (WCCAP).**
2) **Periodically compare measurements from different instruments for consistency in the field. For example, we found that comparing the integrated aerosol volume distribution from the aerosol sizer with the converted total aerosol mass from the AMS measurement can help check both the instrument performances and the inlet operation condition. Additionally, measurements from different cloud probes should be compared in the overlapping size ranges.**
3) **Daily calibration would be valuable but likely unrealistic to perform in the field. One alternative is to daily even hourly monitor the variation of the critical instrument parameters, such as the aerosol sample flow of the individual aerosol instruments.**
4) **For the cases with minor variations in the calibration results, the typical practice is to use the average calibration results for the variation period. However, we also recommend documenting the corresponding uncertainty with the data product."**

Minor comments

Line 373: "exams" should be "examines"

**Response: Corrected in the current Line 370.**

The units of the quoted altitudes change from feet to metres, it would be useful if consistency was maintained throughout. Eg: lines 419 and 420, and elsewhere

**Response: Corrected in the current Line 414.**

Line 457-460 "criterion" and "criteria" singular and plural are used interchangeably

**Response: Corrected in the current Line 447.**

Line 463: are 4dp in the R2 justified?

**Response: Yes, the 10 seconds average is to reduce the variation due to the spatial distance between two aircraft and the wind effect. The increasing of $R^2$ indicates that the spatial variation and wind all contribute to the variation in the airborne measurement.**

Lines 467-468: "the challenging condition of airborne condition"

**Response: Changed to "the challenging condition of airborne measurement" in the current line 457.**

Line 491: different airmass(es)

**Response: Corrected to "different airmasses" in the current line 476.**

Lines 500-502: "Above 3500 m, 500 the variations on the CPC and UHSAS measured concentration significantly smaller than the 501 variation at the lower altitude" needs a verb?

**Response:  Revised to "Above 3500 m, the variations on the CPC and UHSAS measured concentration became significantly smaller than the variation at the lower altitude." In the current line 481.**

Line 621: use super and sub scripts

**Response: Corrected in the current line 530.**

Line 722: no need for the apostrophe

**Response: Corrected in the current line 627.**

Lines 727-728: "the Science engineering Associates" should be "Science Engineering Associates"

**Response: Corrected in the current line 631.**

Line 729: "CDP had placed an 800-µm-diameter pinhole" should be "The CDP had an 800-µm-diameter pinhole placed"

**Response: Corrected in the current line 632.**

Line 730: the FCDP….

**Response: Corrected in the current line 634.**

Line 806-810: I am not sure that this is something that should be concluded in the way it is written. What is said is that there is around a 20% variation between two aircraft flying in essentially the same airmass due to spatial and temporal variability and as a result, modellers should use longer term data to constrain results. True, but realistically, large scale models require a greater degree of averaging than this for a range of other reasons.

**Response: Revised to "Thus, more routine and long-term airborne measurements should be used to evaluate or constrain atmospheric modeling work, especially for the regional or small-scale modeling work." In the current line 703-704.**

Lines 911-912: This needs a rewrite, I am not clear what is being said.

**Response: Revised in the current line 752-754 "The aerosol number concentration and the trace gas measurements both suggest inhomogeneous aerosol distribution between 2000-3000 m altitude."**

Lines 915: aerosol "sample" flow variations

**Response: Corrected in the current line 757.**

Lines 918-919: "Around 70% organic contribution maintains constant up to 3500 m, then decreases to 50%." Change to "Around 70% of the AMS measured mass is organic and this fractional contribution is maintained from the surface to 3500 and decreases to 50% at higher altitudes."

**Response: Corrected in the current line 761-762.**

Line 920: "different mass resolution of the AMS instruments"

**Response: Corrected in the current line 764.**

[revised manuscript text omitted]

---

## Author Response (AR3)

**Associate Editor Decision: Publish subject to minor revisions (review by editor) (18 Dec 2019)**
**by Wiebke Frey**
Comments to the Author:
Dear Fan Mei et al., thanks for your revisions. Following up on these, I have a some more comments, see below.

**Response: We sincerely appreciate the comments and suggestions from our associate editor Wiebke Frey.  Thank you very much for considering the publication of our manuscript. We address your comments below (also in blue). Note that the included line number is based on the mark-up version, which may be different from the final version.**

Main comments:
1) With regard to the last reviewers comments about ground comparisons, I have the impression that you did not fully address the question. From how I understand your answer, comparisons to ground instrumentation had not been carried out, right? Why not? You refer to the ground site a number of times in your manuscript (e.g lines 31-33, lines 66-71, l 83/84, l 563-565, l 581-584) which makes me wonder why there was no comparison of ground based and airborne data. Such comparison might have given an even more comprehensive view on the reliability of the airborne data. Please also add a comment on the (missing?) comparison in the manuscript (and possibly a recommendation in the summary?).
**Response: The last reviewer's comments are on the comparison of the similar instruments deployed on G1 and HALO side by side on the ground, as in a typical setting of a lab-based comparison study. We clarified that we did not perform such a comparison due to logistic difficulties.It would also be beneficial to compare similar instruments deployed at the ground site and those aboard aircraft side by side. Unfortunately, we did not get a chance to do that either.**

**We agree with the editor that a comparison of the aircraft measurements with similar measurements from the ground site (i.e., when aircraft overflew the site) might provide a more comprehensive view of the reliability of the airborne data. However, such a comparison will also be influenced by the spatial variations (both vertical and horizontal). We found that how we define the "flew over" area would affect the comparison results. A comprehensive comparison between the aircraft and ground measurements will go beyond the scope of the current study. We appreciate the excellent suggestion from the editor and added the below recommendation in the summary section (line 803-805). "A side-by-side comparison among the similar instruments deployed at different platforms, including those at ground sites is highly recommended and will provide a comprehensive view of the data reliability."**

2) Numbering:
There are several instances where I find the numbering does not follow the usual order. Starting with the numbering of author affiliations - the third affiliation that appears in the author list is number 13

and not, as I would expect, number 3.

Table S2 and S3 are mentioned in the manuscript before S1. Also some of the figures in the supplement are at higher numbers than others that get mentioned later.

So please check numbering of author affiliations, Figures, and Tables, also in the supplement.

**Response: We have revised the manuscript and the supplement. For example, we revised the order of the author's affiliations and the supplemental tables (in the manuscript line 122).**

3) Supplement:

The supplement reads as if the collection of sections/figures/tables was not always updated after an iteration of the review. Please treat the supplement with the same care as the main manuscript, keep in mind, that the supplement is published along with the final paper.

For some Sections of the supplement, I could not find references to it in the main manuscript. Either include references (e.g. to Section S1) in the main manuscript, or remove from the supplement. Actually, I could only find references to the flight on October 1 (without mentioning the exact position/section in the supplement), Table S2 and S3, and Figures S13 and S14.

**Response: We have revised the content and the figures in the supplement. We removed the section S3 and included more details with the below responses for each specific comment.**

Further comments:

l 115: trace gases concentrations -> trace gas concentrations

**Response: Corrected in the current Line 117.**

l 141: this comparson -> the comparison

**Response: Corrected in the current Line 143.**

l 242: number of small shattered particles (remove brackets)

**Response: Corrected in the current Line 244.**

l 269: developed by Weigel et al. (2016) -> move opening bracket

**Response: Corrected in the current Line 271.**

l 299: The other coordinated flight on October 1 is included in the supplemental document. -> Please include references to specific text/Section and Figures in supplement.

**Response: The reference to Figure S1 and S2 is in line 302 of the main manuscript. The reference to Figure S3-S5 is in line 418 of the main manuscript. The reference to Figure S6 is in line 121 and 130 of the supplemental material. The reference to Figure S7 and S8 is in both lines 302 and 547 of the main manuscript. The reference to Figure S9 is in line 167-168 of the supplemental material. The reference to Figure S10 is in line 172 of the supplemental material.The reference to Figure S11 and S12 is in line 590 of the main manuscript. The**

**reference to Figure S13 is in line 652 of the main manuscript. The reference to Figure S14 is in line 674 of the main manuscript. The reference to section S1 is in line 302. The reference to section S2 is in line 546. The reference to section S3 is in line 566. The reference to the table S1 and S2 is in line 123. The reference to the table S3 is in line 717.**

l 360/361: variation of the G1 altitude flight legs when the G1 flew around 50m/s higher...
Do you mean 'altitude' and 'higher' or 'flight speed' and 'faster'?

**Response: Changed to "faster" in the current Line 364.**

l 362/363: You argue that the G1 has a larger standard deviation in pressure due to the faster flight speed than normal - but HALO flew slower than normal (lines 304/305), would that no also be a possible source for increased errors in HALO observations?

**Response: When the G1 flew at a faster airspeed, it did not maintain the altitude as good as HALO, which flew at a slower airspeed. Thus, the substantial variation of the G1 altitude contributed to the larger standard deviation in the G1 pressure measurements.**

l 374: Two regression results in Table 4 doesn't -> don't
**Response: Changed to "don't" in the current Line 377.**

l 377: Figure 4 - see comment 2)

**Response: Figure 3 is in line 315. Figure 4 is in the current line 381. And figure 5 is in line 411.**

l 385: chilled mirror hydrometer - hygrometer?

**Response: Changed to "hygrometer" in the current Line 389.**

l 395: different saturations: How do you think will saturations impact on wind speed measurements?

**Response: The different saturations during the evaporation-condensation processes will affect the ambient temperature measurement and differential pressure measurements. Then due to the error propagation, it will affect the derived wind speed.**

l 447: The first criterion is the same criterion - maybe repeat it here with a few words.
**Response: Changed to "The first criterion is the same criterion described in the previous section that makes sure all the compared measurements happen in less than 30 minutes apart." in the current line 452-453.**

l 454: Should the references in brackets be in the sentence? Move point back.
**Response: Corrected in the current Line 459.**

l 488: You sometimes say 'cloud-free flight' and sometimes you say the dates of the flights, please be consistent.

**Response: changed to "cloud-free flight on September 9" in the current line 303, 402, 443, and 493.**

l 566: 0.17+-0.06 derived -> space missing

**Response: Corrected in the current Line 571.**

l 582: Figure 3, two aircraft -> both aircraft...

**Response: Changed to "both" in the current line 588.**

l 584: see comment 2)

**Response: Changed to "Figure S11 and Figure S12".**

l 636: You used a correction for coincidences on the FCDP data? In lines 596-598 you explicitly say that coincidence bias should not be considered for the cloud probe measurements to avoid deviations caused by application of different corrections!

**Response: for the G1 flights, we applied the coincidence correction to the final FCDP data product, which the user can access online. However, for the comparison in this paper, we used the FCDP data without applying the coincidence correction. We added "in the final data product" to line 642.**

l 701: This will lead to a considerable uncertainty ... -> add 'to'?

**Response: added "to" in the current Line 708.**

l 702-704: Can you elaborate why "especially for the regional or small scale modelling work"? For example, if someone does a modelling case study of a specific convective cloud, wouldn't more routine or long term measurements smear out the specifics/characteristics of that particular case study storm?
**Response: We try to point out that the modeling evaluation should not be based on the data from a single flight, but the data from more routine and long-term flights. To confirm any physical features in the regional model should require more flights over the region. Changed to "Thus, to evaluate or constrain atmospheric modeling work, more routine and long-term airborne measurements should be used to provide statistically sufficient observation." In the current line 709-712.**

l 730: The UHSAS measurements ... (plural - two instruments being compared), same for CPC measurements.

**Response: Corrected in the current Line 738-739.**

Supplement:

Figures: Please move the Figures to the appropriate text within the Supplement, e.g. Figure 3 to Section S2 - CCN closure.

**Response: We revised the whole document and rearranged the figures.**

l 24: described by Mei et al. (2013b). -> brackets missing

**Response: Corrected in the current Line 165.**

l 24: CCN number concentration (typo)
**Response: Corrected in the current Line 166.**

l 26/27: Figures S3(a) and S3(b) (not Figure 3S...)
**Response: Corrected in the current Line 167 and 168.**

Figures S3 and S4: Please include 1:1 lines for guidance.

**Response: Revised the current figure S9.**

Section S3:
I have to admit, that I am highly skeptical about this section: Extrapolation of aerosol size distributions towards the smaller sizes should in my opinion not be done. The extrapolation is extremely uncertain, as you have to make assumptions about whether sources (particle formation events!) or sinks are present. As I cannot find a link to this section from the main manuscript, I suggest removing this section.

**Response: Removed the aerosol size distribution section.**

l 104: What is "AB, IE, and RIE"?
**Response: Corrected in the current Line 102 -103 .**

l 107-114: Sorry, but this part sounds like copy & paste from an email. Please keep in mind that the supplement will be part of the published manuscript and should be written like such. Please rephrase!
**Response: We are sorry that we sent a wrong version of the supplemental material to Lorena after the first version crashed. We included the revised version in the current line 98 – 114.**

l 117: the inlet flow was kept constant -> keep tense as in the other sentences before/after.
**Response: Corrected in the current Line 117.**

l 134: (Molleker, S., in prep.) -> still in preparation or submitted in meantime?

**Response: Yes, it is still in preparation, but close to submit.**

Figures:

Please check for consistency in axis labelling (N, #/cc vs N, cm-3 for example), and for full axis labelling (some axes do not have labels at all).

**Response: Revised figure S3-S5, S11 and S12.**

S5-7: Occurrence not occurance, why Frequency and not Occurrence in Fig 6a?

**Response: Corrected the current figure S3-S5.**

S11: Would be better to plot similar to S3 (N vs N with altitude as colour code), including 1:1 line.

**Response: Removed the previous S11 with the section S3.**

Table S2 and S3, cloud probes: In the manuscript you state that the G1 cloud probes were calibrated biweekly, in the supplement you write weekly calibration. For HALO cloud probes (e.g. the two CIPgs instruments) you state spinning disk calibrations before or after each flight, again, weekly calibration stated in the table. Please be consistent! What is the real frequency of calibration?

**Response: Corrected the errors in the previous table S2 and S3 (now table S1, S2), and the main manuscript in the current line 123 and 246.**

[revised manuscript text omitted]

**2.Additional information for AMS**

Most of the details for the AMS measurements have been included in the separate AMS papers
(Schulz et al., 2018; Shilling et al., 2018). Brief summaries are provided blew.

The G1 AMS was operated with a constant pressure inlet (CPI), which was set to a constant
pressure during the campaign. The G1 AMS was calibrated  once a week during
the deployment. One additional calibration was performed after the flight
day and  all the calibrations were in agreement with  each other.
Based on five calibrations, the averaged parameters such as the airbeam signal (AB), the
ionization efficiency (IE) and the relative ionization efficiency (RIE)
were applied to all of the data.
A
real-time correction was made to account for the variations in the AB changes  to
improve the instrument sensitivity. Typically, this correction is small (<20%) in absolute
magnitude.

~~and the ~3 hour warmup time we have is long enough for sufficient stability to be achieved, but~~
The particle
collection efficiency (CE) was determined by comparing AMS data to UHSAS and FIMS data.
We also confirmed the CE=0.5 by comparing mass loadings observed at the T3 site to the G1 data.

The HALO- AMS was calibrated before, during (twice) and after the campaign for (relative)
ionization efficiencies -of nitrate, ammonium and sulfate (Schulz et al., 2018). For organics, the
default relative ionization efficiency of 1.4 was assumed. The inlet flow ~i~was kept constant by the
CPI and was measured before and during the campaign. A collection efficiency of 0.5 was applied,
as recommended by Middlebrook et al. (2012) for low nitrate conditions. Further details on the
operation of the C-ToF-AMS are given in Schulz et al. (2018).

Figure S6(a) shows vertical profiles of the total mass concentrations measured by the two
AMS instruments on September 21. Above 2500 m altitude, the agreement between the two
instruments is excellent (mean difference less than 5%). Between 2000 and 2500 m, the agreement
is within the uncertainty range. Below 2000 m altitude, however, the aerosol particle mass
concentrations measured by the AMS operated on HALO are lower than the concentrations
measured by the AMS on the G1. To compare AMS data to UHSAS data, the aerosol mass
concentrations of the G1 AMS were converted to the aerosol volume concentration assuming an
organic compound density of 1.5 g cm$^{-3}$ (Pöschl et al., 2010). The converted aerosol volume
concentration agreed well with the volume concentration calculated based on UHSAS data,
especially below 2500 m, as shown in Figure S6(b). The agreement at lower altitudes suggests that
the lower concentration measured by the HALO AMS is due to the transmission efficiency issue
in the constant pressure inlet used by the HALO AMS. This inlet was a prototype, designed and
built at MPIC Mainz, and works by changing the size of the critical orifice that regulates the flow
into the aerodynamic lens. The design and transmission characteristics will be described in an
upcoming publication (Molleker, S., in prep.). The AMS aboard the G-1 used a constant pressure
inlet based on the design in Bahreini et al., 2008. Thus, we conclude that data above 2500 m
altitude measured by the AMS aboard HALO in 2014 are valid, while data below 2500 m need to
be corrected using correction factors derived from laboratory characterization before further study.

After 2014, the HALO AMS inlet design was improved to address the inlet transmission issues specific to this field campaign.

The second comparison between the two AMS conducted on October 01 is shown in

Figures S6 and S7. The findings are basically in agreement with those of September 21, although the underestimation of aerosol mass concentration due to the inlet in the HALO AMS appears here to be restricted to altitudes lower than 1500 m.

[Figure]

(a)                                                                (b)

Figure S6. (a) Comparison of aerosol mass loading measured by the G1 and HALO AMS on

September 21; (b) aerosol volume concentration comparison from AMS and the integrated

UHSAS on the G1.

[Figure]

Figure S7. The vertical profiling of the aerosol mass concentration observed by the G1 and HALO during October 1.

[Figure]

Figure S8. The vertical profiling of the relative fractions for the chemical species observed by the

G1 and HALO during October 1.

**3. CCN closure**

To further examine the relative importance of mixing state and chemical composition, the CCN concentrations were calculated from aerosol particle size distribution, and chemical composition measured onboard the G1. The calculation was based on $\kappa$-Köhler parameterization, (Kohler, 1936; Petters and Kreidenweis, 2007, 2008, 2013) and the detail of the approach was described by Mei et al. (2013b). For the flight on September 9, 2017, the CCN number concentration calculated from the G1 UHSAS size distribution and chemical composition exhibits underestimation at a supersaturation of 0.5% (Fig. S9(a)) and when the altitude is below 1000 m (Fig S9(b)). This underestimation suggests that the UHSAS size range (90-500 nm) did not fully cover the aerosols with the critical activation diameter ($D_{p,50}$) at high supersaturation. Thus, the FIMS measurements onboard the G1 was the more appropriate size distribution for both the CCN closure study. The CCN concentration calculated using the size distribution from FIMS agrees well with the measurement (Fig. S10). The scattering of the comparison data in Figure 15 is likely due to the chemical composition and mixing state effect on aerosol hygroscopicity.

[Figure]

(a)                                                   (b)

Figure S9. Comparison of calculated CCN with measured CCN using the averaged 1 min measurements from the G1: (a) colored by different supersaturations.  (b) colored by different altitude. (Note that both plots used the calculated CCN number concentration from UHSAS size distribution.)

[Figure]

Figure S10. The scatter plot of the calculated CCN number concentration using FIMS size

  distribution compared with the measured CCN number concentration

[Figure]

Figure S118. The cloud droplet number concentration from the G1 aircraft on September 21.

[Figure]

Figure S129. The cloud droplet number concentration from HALO on September 21.

[Figure]

                                          (a)

[Figure]

(b)

[Figure]

(c)

Figure S13̶0̶. The averaged cloud droplet size distributions from HALO on September 21, (a)
CCP probes; (b) NIXE-CAPS probes; (c) Cloud probes on board the G1.

[Figure]

Figure S14. Time series comparison of the G1 (SPN-1) and HALO (SMART-Albedometer)

          radiation measurements on September 9.

Table S1. Calibration and maintenance for the instruments deployed on G1

| Measurement Variables | Instruments deployed on the G1 (Martin et al., 2016; Schmid et al., 2014) | Calibration/Maintenance |
|---|---|---|
| Static Pressure | Rosemount (1201F1), 0-1400 hPa | Calibrated before/after each field campaign |
| Static air temperature | Rosemount E102AL/510BF -50 to +50 °C | Calibrated before/after each field campaign |
| Dewpoint temperature | Chilled mirror hygrometer 1011B -40 to +50 °C | Calibrated before/after each field campaign |
| 3-D wind | Aircraft Integrated Meteorological Measurement System 20 (AIMMS-20) | Calibrated with Special flight pattern before each field campaign. Inter-comparison with other GPS/INS during deployment. |
| Particle number concentration | CPC, cut off size ($D_p$) =10 nm | Calibrated before/after each field campaign. Weekly calibration of sample and sheath flow rates and inter-comparisons with similar counters during deployment. |
| Size distribution* | UHSAS-A, 60-1000 nm. | Calibrated before/after each field campaign. Weekly check of sizing with PSL |
| FIMS | 10 nm – 500 nm | Calibrated before/after each field campaign. Weekly calibration of sample and sheath flow rates and checks with one size PSL |
| Non-Refractory particle chemical composition | HR-ToF-AMS: Organics, Sulfate, Nitrate, Ammonium, Chloride, 60-1000 nm | Weekly calibrations. |

| CCN concentration | CCN-200, SS= 0.25, 0.5% | Calibrated before/after each field campaign. Biweekly calibration with ammonium sulfate particles. |
| --- | --- | --- |
| Gas phase concentration | N2O/CO and Ozone Analyzer, CO, $O_3$ concentration, precision 2 ppb | Calibrated before/after each field campaign with calibration gas mixture. |
| CDP | 2-50 μm, $\Delta D_P$=1-2 μm | Calibrated before/after each field campaign by the vendor. Weekly check of sizing with glass beads of several sizes |
| FCDP | 2-50 μm, $\Delta D_P$=1-2 μm | Calibrated before/after each field campaign by the vendor. Weekly check of sizing with glass beads of several sizes |
| 2DS | 10-1000 μm | Calibrated before/after each field campaign by the vendor. |
| Radiation | SPN1 downward irradiance, 400-2700 nm | Calibrated before/after each field campaign |

Table S2. Calibration and maintenance for the instruments deployed on HALO

| Measurement Variables | Instruments deployed on HALO (Wendisch et al., 2016) | Calibration/Maintenance |
| --- | --- | --- |
| Static Pressure | Instrumented nose boom tray (DLR development), 0-1400 hPa | Calibrated before/after each field campaign |
| Static air temperature | Total Air Temperature (TAT) inlet (Goodrich/Rosemount type 102) with an open wire resistance temperature sensor (PT100), -70 to +50 °C | Calibrated before/after each field campaign |
| Dewpoint temperature | Derived from the water-vapor mixing ratio, which is measured by a tunable diode laser (TDL) system (DLR development), 5-40000 ppmv | Calibrated before/after each field campaign |
| 3-D wind | Instrumented nose boom tray (DLR development) with an air data probe (Goodrich/Rosemount) 858AJ and high-precision Inertial Reference System (IGI IMU-IIe) | Calibrated before/after each field campaign |
| Particle number concentration | CPC, cut off size ($D_P$) =10 nm | Calibrated before/after each field campaign. Weekly inter-comparisons with similar counters during deployment. |
| Size distribution* | UHSAS-A, 60-1000 nm. | Calibrated before/after each field campaign. Weekly check of sizing with PSL |
| Non-Refractory particle chemical composition | C-ToF-AMS: Organics, Sulfate, Nitrate, Ammonium, Chloride, 60-1000 nm |  Calibrated before and after the campaign and twice during the campaign |
| CCN concentration | CCN-200, SS= 0.13-0.53% | Calibrated before/after each field campaign. Weekly calibration with ammonium sulfate particles. |

Formatted Table

| Gas phase concentration | N2O/CO and Ozone Analyzer, CO, $O_3$ concentration, precision 2 ppb | Calibrated before/after each field campaign with calibration gas mixture. |
|---|---|---|
| Cloud properties* | CCP-CDP, 2.5-46 mm, $\Delta D_P$=1-2 μm | Calibrated before/after each field campaign. Weekly check of sizing with glass beads of several sizes |
| | NIXE-CAS: 0.61 -52.5 μm | Calibrated before/after each field campaign. Weekly check of sizing with glass beads of several sizes |
| | NIXE-CIPgs, 15-960 μm | Calibrated before/after each  flight with a spinning disk. |
| | CCP-CIPgs: 15-960 μm | Calibrated before/after each  flight with a spinning disk. |
| Radiation | SMART Albedometer, downward spectral irradiance, 300-2200 nm | Weekly calibrations. |

Table S3. List of compared measurements ranges and measurement variances caused by the spatial variation during the field campaign.

| Measurement Variables | Measured Range during the Field Campaign | Measurement Variances between the Two Aircraft |
|---|---|---|
| Static Pressure | 500 – 1010 hPa | < 1 % |
| Static air temperature | 272 – 310 K | < 1% |
| Dewpoint temperature | 230 -300 K | Without clouds, <1% With clouds, the measurement from the G1 can be up to 5% lower than that of HALO |
| 3-D wind | 1-15 m/s | < 40% |
| Particle number concentration | 500 – 15,000 $cm^{-3}$ | < 20% for CPC, <50% for UHSAS (size dependent) |
| Non-Refractory particle chemical composition | < 10 μg·$m^{-3}$ | < 10% above 2500 m  Up to 50% below 2500 m |
| CCN concentration | SS=0.25%, 100 – 2000 $cm^{-3}$ | < 10% above 2500 m  Up to 50% below 2500 m |
| Gas phase concentration | Ozone: 15-75 ppb CO: 50-200 ppb | Ozone: < 25% CO: < 15% |
| Cloud droplet number concentration | 3- 20 μm | <50 % |
| Downward irradiance | 200 -1500 W·$m^{-2}$ | < 10% |

---

## Author Response (AR4)

**Associate Editor Decision: Publish subject to technical corrections (16 Jan 2020) by Wiebke Frey**

**Comments to the Author:**

Dear Fan Mei et al., I found some remaining very minor issues in the supplement:

Response: We sincerely appreciate the comments and suggestions from our associate editor Wiebke Frey. Thank you very much for considering the publication of our manuscript. We address your comments below (also in blue).

line 45: 'blew' -> below

**Response: Corrected in the line 46.**

line 50: missing white space: '(RIE)were'

**Response: Corrected in the current line 51.**

line 76: You say G-1 here, but G1 elsewhere, please be consistent.

**Response: Corrected in the current line 77.**

line 82: 'October 01' here, October 1 elsewhere, please be consistent.

**Response: Corrected in the current line 83.**

Sections: Some Figure do not really belong to the sections they appear in, for example in Section 1 Figure S2-S5 are not related to 'Pilot preparation'. Maybe make the 'pilot preparation' part a subsection of a section that could be called something like 'Additional Observations'. Figures S11-S13 are not related to 'CCN closure' and could thus be in a new Section etc.

Response: We changed the section 1 to "Additional observations" and added additional section titles to the related figures: section 4 "Cloud probe observations" (in the current line 135) before the figure S11-S13 and section 5 "Radiation measurements" (in the current line 155) before the figure S14.

For final supplement: Please check that figure captions are not on another page than the respective figure itself!

Response: We revised the figure and captions accordingly.

Best wishes, Wlebke Frey

**1 Supplemental material:**

**2 1. Additional observations**

3 Both aircraft cannot appear at the same location at the same time due to safety concerns. Thus, the approval of a formation (inter-comparison) flight was acquired six months before the 4 campaign through DOE Pacific Northwest Site Office (PNSO) and the Office of Aviation 5 Management (OAM). Essential risk mitigation was also discussed and approved by the Pacific 6 Northwest National Laboratory Aviation Risk Management Committee (PNNL ARMC). During 7 the IOP, both aircraft crew and scientists teams set up a meeting to discuss the potential flight plan. 8 9 After the flight plan was formed, both pilots briefed the plan to the Brazilian Air Force (BAF) and Airport Traffic Control (ATC). The clear-sky flight would be under Visual Flight Rules (VFR), 10 11 which means good weather and no cloud, and pilots communicate with each other using an air-to-12 air frequency. For coordinated flights in cloudy conditions, the G1 and the HALO were both on Instrument Flight Rules (IFR) flight plan. 13

The coordinated flight on October 1, 2014, was initially designed to be a coordinated flight under a cloudy condition, which means the G1 and the HALO flew the same flight leg with at least 300 m altitude offset and at least 5 minutes apart. However, the coordinated two flight legs (~900 m and ~1200 m) are all below the cloud. Thus, the comparison focus on the correlation between two aircraft measurements, not vertical profiling.

19

Figure S1. Time colored flight track of the G1 (circle) and the HALO (triangle) on October 1,
2014, during a cloudless coordinated flight (This figure was created using Mapping ToolboxTM

24 © COPYRIGHT 1997–2019 by The MathWorks, Inc).

Figure S2, Atmospheric parameters observed by the G1 and the HALO on October 1, 2014.